# Don't fear the unlabelled:
# Safe semi-supervised learning via simple debiasing

## Abstract

Semi-supervised learning (SSL) provides an effective means of leveraging un-labelled data to improve a model's performance. Even though the domain has received a considerable amount of attention in the past years, most methods present the common drawback of lacking theoretical guarantees. Our starting point is to notice that the estimate of the risk that most discriminative SSL methods minimise is biased, even asymptotically. This bias impedes the use of standard statistical learning theory and can hurt empirical performance. We propose a simple way of removing the bias. Our debiasing approach is straightforward to implement and applicable to most deep SSL methods. We provide simple theoretical guarantees on the trustworthiness of these modified methods, without having to rely on the strong assumptions on the data distribution that SSL theory usually requires. In particular, we provide generalisation error bounds for the proposed methods. We evaluate debiased versions of different existing SSL methods, such as the Pseudo-label method and Fixmatch, and show that debiasing can compete with classic deep SSL techniques in various settings by providing better calibrated models. Additionally, we provide a theoretical explanation of the intuition of the popular SSL methods.

## 1 Introduction

The promise of semi-supervised learning (SSL) is to be able to learn powerful predictive models using partially labelled data. In turn, this would allow machine learning to be less dependent on the often costly and sometimes dangerously biased task of labelling data. Early SSL approaches—e.g. Scudder's (1965) untaught pattern recognition machine—simply replaced unknown labels with predictions made by some estimate of the predictive model and used the obtained *pseudo-labels* to refine their initial estimate. Other more complex branches of SSL have been explored since, notably using generative models (from McLachlan, 1977, to Kingma et al., 2014) or graphs (notably following Zhu et al., 2003). Deep neural networks, which are state-of-the art supervised predictors, have been trained successfully using SSL. Somewhat surprisingly, the main ingredient of their success is still the notion of pseudo-labels (or one of its variants), combined with systematic use of data augmentation (e.g. Xie et al., 2019; Sohn et al., 2020; Rizve et al., 2021).

An obvious SSL baseline is simply throwing away the unlabelled data. We will call such a baseline the *complete case*, following the missing data literature (e.g. Tsiatis, 2006). As reported in van Engelen & Hoos (2020), the main risk of SSL is the potential degradation caused by the introduction of unlabelled data. Indeed, semi-supervised learning outperforms the complete case baseline only in specific cases (Singh et al., 2008; Schölkopf et al., 2012; Li & Zhou, 2014). This degradation risk for generative models has been analysed in Chapelle et al. (2006, Chapter 4). To overcome this issue, previous works introduced the notion of *safe* semi-supervised learning for techniques which never reduce predictive performance by introducing unlabelled data (Li & Zhou, 2014; Guo et al., 2020). Our loose definition

of safeness is as follows: *an SSL algorithm is safe if it has theoretical guarantees that are similar or stronger to the complete case baseline*. The "theoretical" part of the definition is motivated by the fact that any empirical assessment of generalisation performances of an SSL algorithm is jeopardised by the scarcity of labels. Unfortunately, popular deep SSL techniques generally do not benefit from theoretical guarantees without strong and essentially untestable assumptions on the data distribution (Mey & Loog, 2019) such as the smoothness assumption (small perturbations on the features $x$ do not cause large modification in the labels, $p(y|pert(x)) \approx p(y|x)$) or the cluster assumption (data points are distributed on discrete clusters and points in the same cluster are likely to share the same label).

Most semi-supervised methods rely on these distributional assumptions to ensure performance in entropy minimisation, pseudo-labelling and consistency-based methods. However, no proof is given that guarantees the effectiveness of state-of-the-art methods (Tarvainen & Valpola, 2017; Miyato et al., 2018; Sohn et al., 2020; Pham et al., 2021). To illustrate that SSL requires specific assumptions, we show in a toy example that pseudo-labelling fails at learning. To do so, we draw samples from two uniform distributions with a small overlap. Both supervised and semi-supervised neural networks are trained using the same labelled dataset. While the supervised algorithm learns perfectly the true distribution of $p(1|x)$, the semi-supervised learning methods (both entropy minimisation and pseudo-label) underestimate $p(1|x)$ for $x \in [1, 3]$ (see Figure 1). We also test our proposed method (DeSSL) on this dataset and show that the unbiased version of each SSL technique learns the true distribution accurately. See Appendix A for the results with Entropy Minimisation.

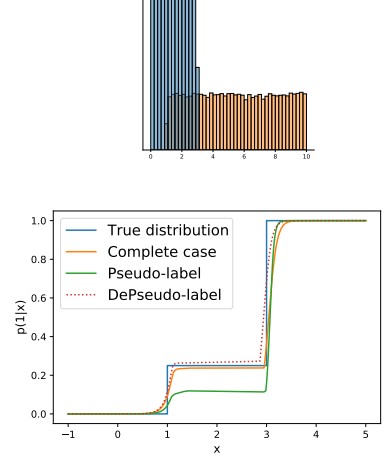

Figure 1: (Left) Data histogram. (Right) Posterior probabilities $p(1|x)$ of the same model trained following either complete case (only labelled data), Pseudo-label or our DePseudo-label.

## 1.1 Contributions

Rather than relying on the strong geometric assumptions usually used in SSL theory, we simply use the *missing completely at random (MCAR)* assumption, a standard assumption from the missing data literature (see e.g. Little & Rubin, 2019). With this only assumption on the data distribution, we propose a new safe SSL method derived from simply debiasing common SSL risk estimates. Our main contributions are:

- We introduce debiased SSL (DeSSL), a safe method that can be applied to most deep SSL algorithms without assumptions on the data distribution;

- We propose a theoretical explanation of the intuition of popular SSL methods. We provide theoretical guarantees on the safeness of using DeSSL both on consistency and calibration of the method. We also provide a generalisation error bound;

- We show how simple it is to apply DeSSL to the most popular methods such as Pseudo-label and Fixmatch, and show empirically that DeSSL leads to models that are never worse than their classical counterparts, generally better calibrated and sometimes much more accurate.

## 2 Semi-supervised learning

### 2.1 Learning with labelled data

The ultimate objective of most of the learning frameworks is to minimise a risk $\mathcal{R}$, defined as the expectation of a particular loss function $L$ over a data distribution $p(x, y)$, on a set of models $f_\theta(x)$, parametrised by $\theta \in \Theta$. Thus, the learning task is finding $\theta^*$ that minimises the risk: $\mathcal{R}(\theta) = \mathbb{E}_{(X,Y) \sim p(x,y)}[L(\theta; X, Y)]$. The distribution $p(x, y)$ being unknown, we generally minimise

an approximation of the risk, the empirical risk $\hat{\mathcal{R}}(\theta)$ computed on a sample of $n$ i.i.d points drawn from $p(x, y)$. $\hat{\mathcal{R}}(\theta)$ is an unbiased and consistent estimate of $\mathcal{R}(\theta)$ under mild assumptions. Its unbiased nature is one of the basic properties that is used for the development of traditional learning theory and asymptotic statistics (van der Vaart, 2000; Shalev-Shwartz & Ben-David, 2014).

## 2.2 Learning with both labelled and unlabelled data

Semi-supervised learning leverages both labelled and unlabelled data to improve the model's performance and generalisation. Further information on the distribution $p(x)$ provides a better understanding of the distributions $p(x, y)$ and also $p(y|x)$. Indeed, $p(x)$ may contain information on $p(y|x)$ (Schölkopf et al., 2012, Goodfellow et al., 2016, Chapter 7.6, van Engelen & Hoos, 2020).

In the following, we have access to $n$ samples drawn from the distribution $p(x, y)$ where some of the labels are missing. We introduce a new random variable $r \in \{0, 1\}$ that governs whether or not a data point is labelled ($r = 0$ missing, $r = 1$ observed). We note $n_l$ the number of labelled and $n_u$ the number of unlabelled datapoints. The MCAR assumption states that the missingness of a label $y$ is independent of its features and the value of the label: $p(x, y, r) = p(x, y)p(r)$, then $r \sim \mathcal{B}(\pi)$. This is the case when nor features nor label carry information about the potential missingness of the labels. This description of semi-supervised learning as a missing data problem has already been done in multiple works –e.g. Seeger, 2000; Ahfock & McLachlan, 2019. Moreover, the MCAR assumption is implicitly made in most of the SSL works to design the experiments, indeed, missing labels are drawn completely as random in datasets such as MNIST, CIFAR or SVHN (Tarvainen & Valpola, 2017; Miyato et al., 2018; Xie et al., 2019; Sohn et al., 2020).

### 2.2.1 Complete case: throwing the unlabelled data away

In missing data theory, the complete case is the learning scheme that only uses fully observed instances, namely labelled data. The natural estimator of the risk is then simply the empirical risk computed on the labelled data. Fortunately, in the MCAR setting, the complete case risk estimate keeps the same good properties of the traditional supervised one: it is unbiased and converges pointwisely to $\mathcal{R}(\theta)$. Therefore, traditional learning theory holds for the complete case under MCAR. While these observations are hardly new (see e.g. Liu & Goldberg, 2020), they can be seen as particular cases of the theory that we develop below. The risk to minimise is

$$\hat{\mathcal{R}}_{CC}(\theta) = \frac{1}{n_l} \sum_{i=1}^{n_l} L(\theta; x_i, y_i). \tag{1}$$

### 2.2.2 Incorporating unlabelled data

A major drawback of the complete case framework is that a lot of data ends up not being exploited. A class of SSL approaches, mainly inductive methods with respect to the taxonomy of van Engelen & Hoos (2020), generally aim to minimise a modified estimator of the risk by including unlabelled data. Therefore, the optimisation problem generally becomes finding $\hat{\theta}$ that minimises the SSL risk,

$$\hat{\mathcal{R}}_{SSL}(\theta) = \frac{1}{n_l} \sum_{i=1}^{n_l} L(\theta; x_i, y_i) + \frac{\lambda}{n_u} \sum_{i=1}^{n_u} H(\theta; x_i). \tag{2}$$

where $H$ is a term that does not depend on the labels and $\lambda$ is a scalar weight which balances the labelled and unlabelled terms. In the literature, $H$ can generally be seen as a surrogate of $L$. Indeed, it looks like the intuitive choices of $H$ are equal or equivalent to a form of expectation of $L$ on a distribution given by the model.

### 2.2.3 Some examples of surrogates

A recent overview of the recent SSL techniques has been proposed by van Engelen & Hoos (2020). In this work, we focus on methods suited for a discriminative probabilistic model $p_\theta(y|x)$ that approximates the conditional $p(y|x)$. We categorised methods into two distinct sections, entropy and consistency-based.

**Entropy-based methods** Entropy-based methods aim to minimise a term of entropy of the predictions computed on unlabelled data. Thus, they encourage the model to be confident on unlabelled data, implicitly using the cluster assumption. Entropy-based methods can all be described as an expectation of $L$ under a distribution $\pi_x$ computed at the datapoint $x$:

$$H(\theta; x) = \mathbb{E}_{\pi_x(\tilde{x}, \tilde{y})}[L(\theta; \tilde{x}, \tilde{y})]. \tag{3}$$

For instance, Grandvalet & Bengio (2004) simply use the Shannon entropy as $H(\theta; x)$ which can be rewritten as equation (3) with $\pi_x(\tilde{x}, \tilde{y}) = \delta_x(\tilde{x})p_\theta(\tilde{y}|\tilde{x})$, where $\delta_x$ is the dirac distribution in $x$. Also, pseudo-label methods, which consist in picking the class with the maximum predicted probability as a pseudo-label for the unlabelled data (Scudder, 1965), can also be described as Equation 3. See Appendix B for complete description of the entropy-based literature (Berthelot et al., 2019; 2020; Xie et al., 2019; Sohn et al., 2020; Rizve et al., 2021; Zhang et al., 2021a) and further details.

**Consistency-based methods** Another range of SSL methods minimise a consistency objective that encourages invariant prediction for perturbations either on the data either on the model in order to enforce stability on model predictions. These methods rely on the smoothness assumption. In this category, we cite $\Pi$-model from (Sajjadi et al., 2016), temporal ensembling from (Laine & Aila, 2017), Mean-teacher proposed by (Tarvainen & Valpola, 2017), virtual adversarial training (VAT) from (Miyato et al., 2018) and interpolation consistent training (ICT) from (Verma et al., 2019). We remark that these objectives $H$ are equivalent to an expectation of $L$ (see Appendix B). The general form of the unsupervised objective can be written as

$$C_1 \mathbb{E}_{\pi_x(\tilde{x}, \tilde{y})}[L(\theta; \tilde{x}, \tilde{y})] \leq H(\theta; x) = \mathbf{Div}(f_{\hat{\theta}}(x, .), pert(f_\theta(x, .)) \leq C_2 \mathbb{E}_{\pi_x(\tilde{x}, \tilde{y})}[L(\theta; \tilde{x}, \tilde{y})], \tag{4}$$

where $f_{\hat{\theta}}$ is the predictions of the model, the $\mathbf{Div}$ is a non-negative function that measures the divergence between two distributions, $\hat{\theta}$ is a fixed copy of the current parameter $\theta$ (the gradient is not propagated through $\hat{\theta}$), *pert* is a perturbation applied to the model or the data and $0 \leq C_1 \leq C_2$.

Previous works also remarked that $H$ is an expectation of $L$ for entropy-minimisation and pseudo-label (Zhu et al., 2022; Aminian et al., 2022). We describe a more general framework covering further methods and provide with our theory an intuition on the choice of $H$.

### 2.3 Theoretical guarantees

The main risk of SSL is the potential degradation caused by the introduction of unlabelled data when distributional assumptions are not satisfied (Singh et al., 2008; Schölkopf et al., 2012; Li & Zhou, 2014), specifically in settings where the MCAR assumption does not hold anymore (Oliver et al., 2018; Guo et al., 2020). Additionally, in (Zhu et al., 2022), the authors show disparate impacts of pseudo-labelling on the different sub-classes of the population. To mitigate these problems, previous works introduced the notion *safe* semi-supervised learning for techniques which never reduce learning performance by introducing unlabelled data (Li & Zhou, 2014; Kawakita & Takeuchi, 2014; Li et al., 2016; Gan et al., 2017; Trapp et al., 2017; Guo et al., 2020). As remark by Oliver et al. (2018), SSL performances are enabled by leveraging large validation sets which is not suited for real-world applications. Then, theoretical guarantees are required to use safely SSL algorithms. For this reason, in our work, we consider as *safe* an SSL algorithm that has theoretical guarantees that are similar or stronger than those of the complete case baseline. Even though the methods presented above produce good performances in a variety of SSL benchmarks, they generally do not benefit from theoretical guarantees, even elementary. More over, Schölkopf et al. (2012) identify settings on the causal relation between the features $x$ and the target $y$ where SSL may systematically fail, even if classic SSL assumptions hold. Our example of Figure 1 also shows that classic SSL may fail to generalise in a very benign setting with a large number of labelled data.

Presented methods minimise a biased version of the risk under the MCAR assumption and therefore classical learning theory cannot be applied anymore, as we argue more precisely in Appendix C. Learning over a biased estimate of the risk is not necessarily unsafe but it is difficult to provide theoretical guarantees on such methods even if some works try to do so with strong assumptions on the data distribution (Mey & Loog 2019, Section 4 and 5). Additionally, we remark that the choice of $H$ can be confusing as seen in the literature. For instance, Grandvalet & Bengio (2004) and Corduneanu & Jaakkola (2003) perform respectively entropy and mutual information *minimisation* whereas Pereyra et al. (2017) and Krause et al. (2010) perform *maximisation* of the same quantities.

## 2.4 Related works

Previous works already proposed safe SSL methods with theoretical guarantees. Unfortunately, so far these methods come with either strong assumptions or important computational burdens. Li & Zhou (2014) introduced a safe semi-supervised SVM and showed that the accuracy of their method is never worse than SVMs trained with only labelled data with the assumption that the true model is accessible. However, if the distributional assumptions are not satisfied, no improvement or degeneration is expected. Sakai et al. (2017) proposed an unbiased estimate of the risk for binary classification by including unlabelled data. The key idea is to use unlabelled data to better evaluate on the one hand the risk of positive class samples and on the other the risk of negative samples. They provided theoretical guarantees on its variance and a generalisation error bound. The method is designed only for binary classification and has not been tested in a deep learning setting. It has been extended to ordinal regression in follow-up work (Tsuchiya et al., 2021). In the context of kernel machines, Liu & Goldberg (2020) used an unbiased estimate of risk, like ours, for a specific choice of $H$. Guo et al. (2020) proposed $DS^3L$, a safe method that needs to approximately solve a bi-level optimisation problem. In particular, the method is designed for a different setting, not under the MCAR assumption, where there is a class mismatch between labelled and unlabelled data. The resolution of the optimisation problem provides a solution not worse than the complete case but comes with approximations. They provide a generalisation error bound. Also, the method does not outperform classic SSL methods in the MCAR setting as it is designed for non-MCAR situations. Sokolovska et al. (2008) proposed a safe method with strong assumptions such that the feature space is finite and the marginal probability distribution of $x$ is fully known. Fox-Roberts & Rosten (2014) proposed an unbiased estimator in the generative setting applicable to a large range of models and they prove that this estimator has a lower variance than the one of the complete case.

## 3 DeSSL: Unbiased semi-supervised learning

To overcome the issues introduced by the second term in the approximation of the risk for the semi-supervised learning approach, we propose DeSSL, an unbiased version of the SSL estimator using labelled data to annul the bias. The idea here is to retrieve the properties of classical learning theory. Fortunately, we will see that the proposed method can eventually have better properties than the complete case, in particular with regard to the variance of the estimate. The proposed DeSSL objective is

$$\hat{\mathcal{R}}_{DeSSL}(\theta) = \frac{1}{n_l} \sum_{i=1}^{n_l} L(\theta; x_i, y_i) + \frac{\lambda}{n_u} \sum_{i=1}^{n_u} H(\theta; x_i) - \frac{\lambda}{n_l} \sum_{i=1}^{n_l} H(\theta; x_i). \tag{5}$$

Under the MCAR assumption, this estimator is unbiased for any value of the parameter $\lambda$. For proof of this result see Appendix D. We prove the optimality of debiasing with the labelled data in Appendix F.

Intuitively, for entropy-based methods, $H$ should be applied only on unlabelled data to enforce the confidence of the model only on unlabelled datapoints. Whereas, for consistency-based methods, $H$ can be applied to any subset of data points. Our theory and proposed method remain the same whether $H$ is applied to all the available data or not (see Appendix K).

### 3.1 Does the DeSSL risk estimator make sense?

The most intuitive interpretation is that by debiasing the risk estimator, we get back to the basics of learning theory. This way of debiasing is closely related to the method of control variates (Owen, 2013, Chapter 8) which is a common variance reduction technique. The idea is to add an additional term to a Monte-Carlo estimator with a null expectation in order to reduce the variance of the estimator without modifying the expectation. Here, DeSSL can also be interpreted as a control variate on the risk's gradient itself and should improve the optimisation scheme. This idea is close to the optimisation schemes introduced by Johnson & Zhang (2013) and Defazio et al. (2014) which reduce the variance of the gradients' estimate to improve optimisation performance.

Another interesting way to interpret DeSSL is as a constrained optimisation problem. Indeed, minimising $\hat{\mathcal{R}}_{DeSSL}$ is equivalent to minimising the Lagrangian of the following optimisation problem:

$$\min_{\theta} \quad \hat{\mathcal{R}}_{CC}(\theta)$$

$$\text{s.t.} \quad \frac{1}{n_u} \sum_{i=1}^{n_u} H(\theta; x_i) = \frac{1}{n_l} \sum_{i=1}^{n_l} H(\theta; x_i). \tag{6}$$

The idea of this optimisation problem is to minimise the complete case risk estimator by assessing that some properties represented by $H$ are on average equal for the labelled data and the unlabelled data. For example, if we consider entropy-minimisation, this program encourages the model to have the same confidence on the unlabelled examples as on the labelled ones.

The debiasing term of our objective will penalise the confidence of the model on the labelled data. Pereyra et al. (2017) show that penalising the entropy in a supervised context acts as a strong regulator for supervised models and improves on the state-of-the-art on common benchmarks. This comforts us in the idea of debiasing using labelled data in the case of entropy-minimisation. Similarly, the debiasing term in pseudo-label turns the problem into plausibility inference as described by Barndorff-Nielsen (1976). Our objective also resembles doubly-robust risk estimates used for SSL in the context of kernel machines by Liu & Goldberg (2020) and for deep learning in a recent preprint (Hu et al., 2022). In both cases, their focus is quite different, as they consider weaker conditions than MCAR, but very specific choices of $H$.

### 3.2 Is $\hat{\mathcal{R}}_{DeSSL}(\theta)$ an accurate risk estimate?

Because of the connections between our debiased estimate and variance reduction techniques, we have a natural interest in the variance of the estimate. Having a lower-variance estimate of the risk would mean estimating it more accurately, leading to better models. Similarly to traditional control variates (Owen, 2013), the variance can be computed, and optimised in $\lambda$:

**Theorem 3.1.** *The function $\lambda \mapsto \mathbb{V}(\hat{\mathcal{R}}_{DeSSL}(\theta))$ reaches its minimum for:*

$$\lambda_{opt} = \frac{n_u}{n} \frac{\text{Cov}(L(\theta; x, y), H(\theta; x))}{\mathbb{V}(H(\theta; x))}, \tag{7}$$

*and at $\lambda_{opt}$:*

$$\mathbb{V}(\hat{\mathcal{R}}_{DeSSL}(\theta))|_{\lambda_{opt}} = \left(1 - \frac{n_u}{n} \rho_{L,H}^2\right) \mathbb{V}(\hat{\mathcal{R}}_{CC}(\theta)) \leq \mathbb{V}(\hat{\mathcal{R}}_{CC}(\theta)), \tag{8}$$

*where $\rho_{L,H} = \text{Corr}(L(\theta; x, y), H(\theta; x))$.*

Additionally, $\mathbb{V}(\hat{\mathcal{R}}_{DeSSL}(\theta)) \leq \mathbb{V}(\hat{\mathcal{R}}_{CC}(\theta))$ for all $\lambda$ between 0 and $2\lambda_{opt}$. A proof of this theorem is available as Appendix E. This theorem provides a formal justification to the heuristic idea that $H$ should be a surrogate of $L$. Indeed, DeSSL is a more accurate risk estimate when $H$ is strongly positively correlated with $L$, which is likely to be the case when $H$ is equal or equivalent to an expectation of $L$. Then, choosing $\lambda$ positive is a coherent choice. We also demonstrate in Appendix E that $L$ and $H$ are positively correlated when $L$ is the negative likelihood and $H$ is the entropy. Other SSL methods have variance reduction guarantees and already has shown great promises in SSL, see Fox-Roberts & Rosten (2014) and Sakai et al. (2017). In a purely supervised context, Chen et al. (2020) show that the effectiveness of data augmentation techniques lays partially on the variance reduction of the risk estimate. A natural application of this theorem would be to tune $\lambda$ automatically by estimating $\lambda_{opt}$. In our case however, the estimation of $\text{Cov}(L(\theta; x, y), H(\theta; x))$ with few labels led to extremely unstable unsatisfactory results. However, we estimate it more accurately using the test set (which is of course impossible in practice) on different datasets and methods to provide intuition on the order of $\lambda_{opt}$ and the range of the variance reduction regime in Appendix M.2.

### 3.3 Calibration

The calibration of a model is its capacity of predicting probability estimates that are representative of the true distribution. This property is determinant in real-world application when we need

reliable predictions. A scoring rule $\mathcal{S}$ is a function assigning a score to the predictive distribution $p_\theta(y|x)$ relative to the event $y|x \sim p(y|x)$, $\mathcal{S}(p_\theta, (x,y))$, where $p(x,y)$ is the true distribution (see e.g. Gneiting & Raftery, 2007). A scoring rule measures both the accuracy and the quality of predictive uncertainty, meaning that better calibration is rewarded. The expected scoring rule is defined as $\mathcal{S}(p_\theta, p) = \mathbb{E}_p[\mathcal{S}(p_\theta, (x,y))]$. A proper scoring rule is defined as a scoring rule such that $\mathcal{S}(p_\theta, p) \leq \mathcal{S}(p, p)$ (Gneiting & Raftery, 2007). The motivation behind having proper scoring rules comes from the following: suppose that the true data distribution $p$ is accessible by our set of models. Then, the scoring rule encourages to predict $p_\theta = p$. The opposite of a proper scoring rule can then be used to train a model to encourage the calibration of predictive uncertainty: $L(\theta; x, y) = -\mathcal{S}(p_\theta, (x,y))$. Most common losses used to train models are proper scorings rule such as log-likelihood.

**Theorem 3.2.** *If $\mathcal{S}(p_\theta, (x,y)) = -L(\theta; x, y)$ is a proper scoring rule, then $\mathcal{S}'(p_\theta, (x,y,r)) = -(\frac{rn}{n_l}L(\theta; x, y) + \lambda n(\frac{1-r}{n_u} - \frac{r}{n_l})H(\theta; x))$ is also a proper scoring rule.*

The proof is available in Appendix G, and follows directly from unbiasedness and the MCAR assumption. The main interpretation of this theorem is that we can expect DeSSL to be as well-calibrated as the complete case.

## 3.4 Consistency

We say that $\hat{\theta}$ is consistent if $d(\hat{\theta}, \theta^*) \xrightarrow{p} 0$ when $n \to \infty$, where $d$ is a distance on $\Theta$. The asymptotic properties of $\hat{\theta}$ depend on the behaviours of the functions $L$ and $H$. We will thus require the following standard assumptions.

**Assumption 3.3.** The minimum $\theta^*$ of $\mathcal{R}$ is well-separated: $\inf_{\theta:d(\theta^*,\theta)\geq\epsilon} \mathcal{R}(\theta) > \mathcal{R}(\theta^*)$.

**Assumption 3.4.** The uniform weak law of large number holds for both $L$ and $H$.

**Theorem 3.5.** *Under the MCAR assumption, Assumption 3.3 and Assumption 3.4, $\hat{\theta} = \arg\min \hat{\mathcal{R}}_{DeSSL}$ is consistent.*

For proof of this theorem see Appendix G. This theorem is a simple application of van der Vaart's (2000) Theorem 5.7 proving the consistency of an M-estimator. Also, this result holds for the complete case, with $\lambda = 0$ which proves that the complete case is a solid baseline under the MCAR assumption. Going further, we prove the asymptotic normality of $\hat{\theta}_{DeSSL}$ and showed that the asymptotic variance can be optimised with respect to $\lambda$.

**Coupling of $n_l$ and $n_u$ under the MCAR assumption** Under the MCAR assumption, $n_l$ and $n_u$ are random variables. We have that $r \sim \mathcal{B}(\pi)$ (i.e. any $x$ has the probability $\pi$ of being labelled). Then, with $n$ growing to infinity, we have $\frac{n_l}{n} = \frac{n_l}{n_l+n_u} \to \pi$. Therefore, both $n_l$ and $n_u$ grow to infinity and $\frac{n_l}{n_u} \to \frac{\pi-1}{\pi}$. This implies $n_u = \mathcal{O}(n_l)$ and then when $n$ goes to infinity, both $n_u$ and $n_l$ go to infinity too and even if $n_u >> n_l$.

## 3.5 Rademacher complexity and generalisation bounds

In this section, we prove an upper bound for the generalisation error of DeSSL. The unbiasedness of $\hat{\mathcal{R}}_{DeSSL}$ can directly be used to derive generalisation bounds based on the Rademacher complexity (Bartlett & Mendelson, 2002), defined in our case as

$$R_n = \mathbb{E}_{(\varepsilon_i)_{i \leq n}} \left[ \sup_{\theta \in \Theta} \left( \frac{1}{n_l} \sum_{i=1}^{n_l} \varepsilon_i L(\theta; x_i, y_i) - \frac{\lambda}{n_l} \sum_{i=1}^{n_l} \varepsilon_i H(\theta; x_i) + \frac{\lambda}{n_u} \sum_{i=1}^{n_u} \varepsilon_i H(\theta; x_i) \right) \right], \quad (9)$$

where $\varepsilon_i$ are i.i.d. Rademacher variables independent of the data. In the particular case of $\lambda = 0$, we recover the standard Rademacher complexity of the complete case. We can then now bound the generalisation error of a model trained using our new loss function.

**Theorem 3.6.** *We assume that labels are MCAR and that both $L$ and $H$ are bounded. Then, there exists a constant $\kappa > 0$, that depends on $\lambda$, $L$, $H$, and the ratio of observed labels, such that, with probability at least $1 - \delta$, for all $\theta \in \Theta$,*

$$\mathcal{R}(\theta) \leq \hat{\mathcal{R}}_{DeSSL}(\theta) + 2R_n + \kappa\sqrt{\frac{\log(4/\delta)}{n}}. \quad (10)$$

310   The proof follows Shalev-Shwartz & Ben-David (2014, Chapter 26), and is available in Appendix J.

# 4   Experiments

312   We evaluate the performance of DeSSL against different classic methods. The goal here is to compare
313   DeSSL methods and their original counterparts. In particular, we perform experiments with simple
314   SSL methods such as pseudo-label (PseudoLabel) and entropy minimisation (EntMIN) with varying
315   $\lambda$ on MNIST (LeCun & Cortes, 2010) and CIFAR-10 and CIFAR-100 (Krizhevsky, 2009) and
316   compare them to the debiased method, respectively DeEntMin and DePseudoLabel. We also compare
317   PseudoLabel and DePseudoLabel on five small datasets of MedMNIST (Yang et al., 2021a;b) with a
318   fixed $\lambda$. The results of these experiments are reported below. In our figures, the error bars represent
319   the size of the 95% confidence interval (CI). Finally, we modified the implementation of Fixmatch
320   (Sohn et al., 2020) and compare it with its debiased version on CIFAR-10.

321   We also compare DeEntMin and DePseudoLabel to the biased version on a large range of tabular
322   datasets commonly used in SSL benchmarks (Chapelle et al., 2006; Guo et al., 2010). We do not
323   observe differences between the performance, see Appendix P. Finally, we show how simple it is to
324   debias an existing implementation, by demonstrating it on the consistency-based models benchmarked
325   by (Oliver et al., 2018), namely VAT, $\Pi$-model and MeanTeacher on CIFAR-10 and SVHN (Netzer
326   et al., 2011). We observe similar performances between the debiased and biased versions for the differ-
327   ent methods, both in terms of cross-entropy and accuracy. Moreover, these results have been obtained
328   using the hyperparameters finetuned for the biased versions. Therefore, it is likely that optimising the
329   hyperparameters for DeSSL will yield even better with the right hyperparameters, see Appendix O.

## 4.1   MNIST

331   MNIST is an advantageous dataset for SSL since classes are
332   well-separated. We compare PseudoLabel and DePseudoLabel
333   for a LeNet-like architecture using $n_l = 1000$ labelled data on
334   10 different splits of the training dataset into a labelled and unla-
335   belled set. Models are then evaluated using the standard $10,000$
336   test samples. We used 10% of $n_l$ as the validation set. We test
337   the influence of the hyperparameter $\lambda$ and report the accuracy,
338   the cross-entropy and the expected calibration error (ECE, Guo
339   et al., 2017) at the epoch of best validation accuracy, see Fig-
340   ure 2 and Appendix L. In this example SSL and DeSSL have
341   almost the same accuracy for all $\lambda$, however, DeSSL seems to
342   be always better calibrated. To break the cluster assumption, we
343   reproduced the same experiment on a modified MNIST. Indeed,
344   we had label noise by replacing the true label for 20% of the
345   dataset with a randomly sampled label, see Appendix L. In this
346   setting, DeSSL performs better for large $\lambda$ in terms of accuracy
347   and also provides a better calibration.

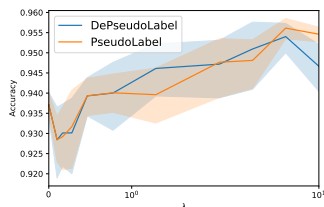

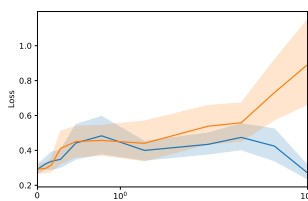

Figure 2: The influence of $\lambda$ on
Pseudo-label and DePseudo-label for
a Lenet trained on MNIST with $n_l =$
$1000$: (Top) Mean test accuracy; (Bot-
tom) Mean test cross-entropy, with
95% CI.

## 4.2   MedMNIST

349   We compare PseudoLabel and DePseudoLabel on different datasets of MedMNIST, a large-scale
350   MNIST-like collection of biomedical images. We selected the five smallest 2D datasets of the
351   collection, for these datasets it is likely that the cluster assumption no longer holds. We trained
352   a 5-layer CNN with a fixed $\lambda = 1$ and $n_l$ at 10% of the training data. We report in Table 1 the
353   mean accuracy and cross-entropy on 5 different splits of the labelled and unlabelled data and the
354   number of labelled data used. We report the AUC in Appendix L. DePseudoLabel competes with
355   PseudoLabel in terms of accuracy and even success when PseudoLabel's accuracy is less than the
356   complete case. Moreover, DePseudoLabel is always better in terms of cross-entropy, so calibration,
357   whereas PseudoLabel is always worse than the complete case.

Table 1: Test accuracy and cross-entropy of Complete Case (CC), PseudoLabel (PL) and DePseudoLabel (DePL) on five datasets of MedMNIST.

| DATASET | NL | CC | | PL | | DEPL | |
|---------|-----|---------------|--------------|---------------|--------------|---------------|--------------|
| | | CROSS-ENTROPY | ACCURACY | CROSS-ENTROPY | ACCURACY | CROSS-ENTROPY | ACCURACY |
| DERMA | 1000 | $1.95 \pm 0.09$ | $68.99 \pm 1.20$ | $2.51 \pm 0.20$ | $68.88 \pm 1.03$ | $\mathbf{1.88 \pm 0.12}$ | $\mathbf{69.30 \pm 0.85}$ |
| PNEUMONIA | 585 | $1.47 \pm 0.04$ | $83.94 \pm 2.40$ | $2.04 \pm 0.04$ | $\mathbf{85.83 \pm 2.13}$ | $\mathbf{1.40 \pm 0.06}$ | $84.36 \pm 3.79$ |
| RETINA | 160 | $1.68 \pm 0.03$ | $48.30 \pm 3.06$ | $1.80 \pm 0.18$ | $47.75 \pm 2.50$ | $\mathbf{1.67 \pm 0.06}$ | $\mathbf{49.40 \pm 2.62}$ |
| BREAST | 78 | $0.80 \pm 0.04$ | $76.15 \pm 0.75$ | $1.00 \pm 0.26$ | $74.74 \pm 1.04$ | $\mathbf{0.70 \pm 0.03}$ | $\mathbf{76.67 \pm 1.32}$ |
| BLOOD | 1700 | $\mathbf{6.11 \pm 0.17}$ | $\mathbf{84.13 \pm 0.83}$ | $6.61 \pm 0.22$ | $84.09 \pm 1.17$ | $6.53 \pm 0.30$ | $83.68 \pm 0.59$ |

## 4.3 CIFAR

We compare PseudoLabel and DePseudoLabel on CIFAR-10 and CIFAR-100. We trained a CNN-13 from Tarvainen & Valpola (2017) on 5 different splits. For this experiment, we use $n_l = 4000$ and use the rest of the dataset as unlabelled. Models are then evaluated using the standard $10,000$ test samples. For a more realistic validation set, we used 10% of $n_l$ as the validation set. We test the influence of the hyperparameter $\lambda$ and report the accuracy and the cross-entropy at the epoch of best validation accuracy, see Figure 3. We report the ECE in Appendix M. The performance of both methods on CIFAR-100 with $nl = 10000$ are reported in Appendix M. We observe DeSSL provides both a better cross-entropy and ECE with the same accuracy for small $\lambda$. For larger $\lambda$, DeSSL performs better in all the reported metrics. We performed a paired Student's t-test to ensure that our results are significant and reported the p-values in Appendix M. The p-values indicate that for $\lambda$ close to 10, DeSSL is often significantly better in all the metrics. Moreover, DeSSL for large $\lambda$ provides a better cross-entropy and ECE than the complete case whereas SSL never does.

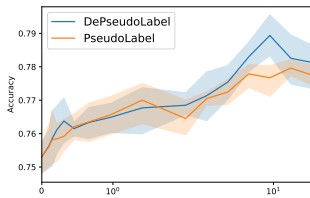

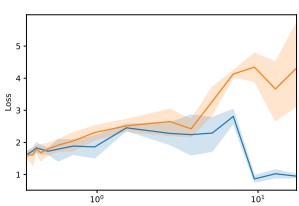

Figure 3: Influence of $\lambda$ on Pseudolabel and DePseudo-label for a CNN trained on CIFAR with $n_l = 4000$: (Left) Mean test accuracy; (Right) Mean test cross-entropy, with 95% CI.

## 4.4 Fixmatch (Sohn et al., 2020)

We debiased a version of Fixmatch, see Appendix N for further details. For this experiment, we use $nl = 4000$ on 5 different folds. First, we report that a strong baseline using data augmentation reach $87.27\%$ accuracy. Then, we observe that on the debiasing method improve both accuracy and cross-entropy of this modified version of Fixmatch. Inspired by Zhu et al. (2022), we show that our method improved performance on "poor" classes more equally than the biased version. Indeed, DeFixmatch improves Fixmatch by $1.57\%$ overall but by $4.91\%$ on the worst class. We report in Appendix N the accuracy per class of the different methods and the *benefit ratio* as defined by Zhu et al. (2022).

Table 2: 1st line: Accuracy, 2nd line: Worst class accuracy, 3rd line: Cross-entropy.

| COMPLETE CASE | FIXMATCH | DEFIXMATCH |
|---------------|----------|------------|
| $87.27 \pm 0.25$ | $93.87 \pm 0.13$ | $\mathbf{95.44 \pm 0.10}$ |
| $70.08 \pm 0.93$ | $82.25 \pm 2.27$ | $\mathbf{87.16 \pm 0.46}$ |
| $0.60 \pm 0.01$ | $0.27 \pm 0.01$ | $\mathbf{0.20 \pm 0.01}$ |

## 5 Conclusion

Motivated by the remarks of van Engelen & Hoos (2020) and Oliver et al. (2018) on the missingness of theoretical guarantees in SSL, we proposed a simple modification of SSL frameworks. We consider frameworks based on the inclusion of unlabelled data in the computation of the risk estimator and debias them using labelled data. We show theoretically that this debiasing comes with several theoretical guarantees. We demonstrate these theoretical results experimentally on several common SSL datasets and some more challenging ones such as MNIST with label noise. DeSSL shows competitive performance in terms of accuracy compared to its biased version but improves significantly the calibration. There are several future directions open to us. We showed that $\lambda_{opt}$ exists (Theorem 3.1) and therefore our formula provides guidelines for the optimisation of $\lambda$. Finally, an interesting improvement would be to go beyond the MCAR assumption by considering settings with a distribution mismatch between labelled and unlabelled data (Guo et al., 2020; Cao et al., 2021; Hu et al., 2022).

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

# A    Toy example

We trained a 4-layer neural network (1/20/100/20/1) with ReLU activation function using $25,000$ labelled and $25,000$ unlabelled points drawn from two 1D uniform laws with an overlap. We used $\lambda = 1$ and a confidence threshold for Pseudo-label $\tau = 0.70$. We optimised the model's weights using a stochastic gradient descent (SGD) optimiser with a learning rate of $0.1$.

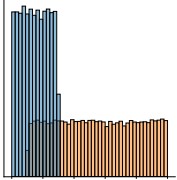

Figure 4: Data histogram

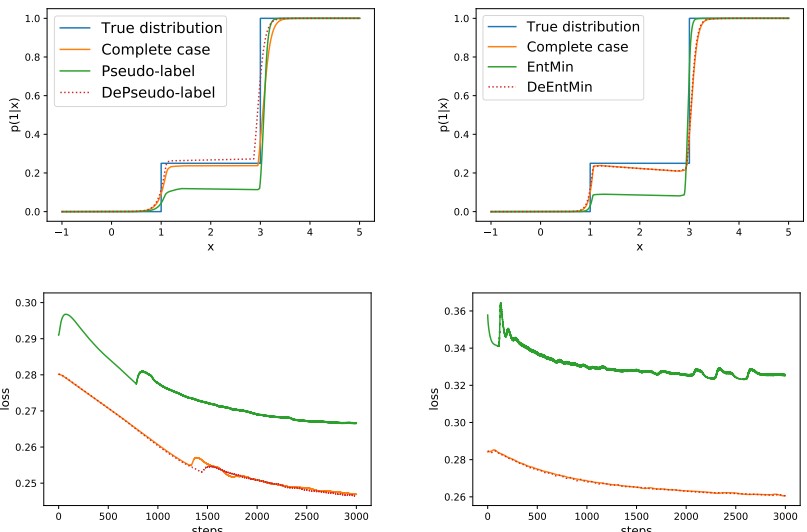

Figure 5: 4-layer neural net trained using SSL methods on a 1D dataset drawn from two uniform laws. (Top-left) Posterior probabilities $p(1|x)$ of the same model trained following either complete case (only labelled data), Pseudo-label or our DePseudo-label. (Top-right) Same for EntMin and DeEntMin (Bottom-left) Training cross-entropy for Pseudo-label and DePseudo-label (Bottom-right) Training cross-entropy for EntMin and DeEntMin

# B  Details on surrogates and more examples

We provide in this appendix further details on our classification of SSL methods between entropy-based and consistency-based (see Section 2.2.3). We detail a general framework for both of these methods' classes. We also show how popular SSL methods are related to our framework.

## B.1  Entropy-based

We class as entropy-based, methods that aim to minimise a term of entropy such as Grandvalet & Bengio (2004) which minimises Shannon's entropy or pseudo-label which is a form of entropy, see Remark E.5. These methods encourage the model to be confident on unlabelled data, implicitly using the cluster assumption. We recall those entropy-based methods can all be described as an expectation of $L$ under a distribution $\pi_x$ computed at the datapoint $x$:

$$H(\theta; x) = \mathbb{E}_{\pi_x(\tilde{x}, \tilde{y})}[L(\theta; \tilde{x}, \tilde{y})]. \tag{11}$$

**Pseudo-label:**  As presented in the core article, the unsupervised objective of pseudo-label can be written as an expectation of $L$ on the distribution $\pi_x(\tilde{x}, \tilde{y}) = \delta_x(\tilde{x})p_\theta(\tilde{y}|\tilde{x})$. Recently, Lee (2013) encouraged the pseudo-labels method for deep semi-supervised learning. Then, Rizve et al. (2021) recently improved the pseudo-label selection by introducing an uncertainty-aware mechanism on the confidence of the model concerning the predicted probabilities. Pham et al. (2021) reaches state-of-the-art on the Imagenet challenge using pseudo-labels on a large dataset of additional images.

## B.2  Pseudo-label and data augmentation

Recently, several methods based on data augmentation have been proposed and proven to perform well on a large spectrum of SSL tasks. The idea is to have a model resilient to strong data-augmentation of the input (Berthelot et al., 2019; 2020; Sohn et al., 2020; Xie et al., 2019; Zhang et al., 2021a). These methods rely both on the cluster assumption and the smoothness assumption and are at the border between entropy-based and consistency-based methods. The idea is to have the same prediction for an input and an augmented version of it. For instance, in Sohn et al. (2020), we first compute pseudo-labels predicted using a weakly-augmented version of $x$ (flip-and-shift data augmentation) and then minimise the likelihood with the predictions of the model on a strongly augmented version of $x$. In Xie et al. (2019), the method is a little bit different as we minimise the cross entropy between the prediction of the model on $x$ and the predictions of an augmented version. In both cases, the unsupervised part of the risk estimator can be reformulated as Equation 11.

**Fixmatch:**  In Fixmatch, Sohn et al. (2020), the unsupervised objective can be written as:

$$H(\theta; x) = \mathbb{1}[\max_y p_{\hat{\theta}}(y|x_1) > \tau]L(\theta; x_2, \arg\max_y p_{\hat{\theta}}(y|x_1)) \tag{12}$$

where $\hat{\theta}$ is a fixed copy of the current parameters $\theta$ indicating that the gradient is not propagated through it, $x_1$ is a weakly-augmented version of $x$ and $x_2$ a strongly-augmented one. Therefore, we write $H$ as an expectation of $L$ on the distribution $\pi_x(\tilde{x}, \tilde{y}) = \delta_{x_2}(\tilde{x})\delta_{\arg\max_y p_{\hat{\theta}}(y|x_1)}(\tilde{y})\mathbb{1}[\max_y p_{\hat{\theta}}(y|x_1) > \tau]$.

**UDA:**  In UDA, Xie et al. (2019), the unsupervised objective can be written as:

$$H(\theta; x) = \sum_y p_{\hat{\theta}}(y|x)L(\theta; x_1, y) \tag{13}$$

where $\hat{\theta}$ is a fixed copy of the current parameters $\theta$ indicating that the gradient is not propagated through it and $x_1$ is an augmented version of $x$. Therefore, we write $H$ as an expectation of $L$ on the distribution $\pi_x(\tilde{x}, \tilde{y}) = \delta_{x_1}(\tilde{x})p_{\hat{\theta}}(\tilde{y}|\tilde{x})$.

**Others:** Recently, have been proposed in the literature Zhang et al. (2021a) and Rizve et al. (2021). The former is an improved version of Fixmatch with a variable threshold $\tau$ with respect to the class and the training stage. The latter introduces a measurement of uncertainty in the pseudo-labelling step to improve the selection. They also introduce negative pseudo-labels to improve the single-label classification.

### B.3 Consistency-based

Consistency-based methods aim to smooth the decision function of the models or have more stable predictions. These objectives $H$ are not directly a form of expectation of $L$ but are equivalent to an expectation of $L$. For all the following methods we can write the unsupervised objective $H$ such that:

$$C_1 \mathbb{E}_{\pi_x(\tilde{x},\tilde{y})}[L(\theta;\tilde{x},\tilde{y})] \leq H(\theta;x) \leq C_2 \mathbb{E}_{\pi_x(\tilde{x},\tilde{y})}[L(\theta;\tilde{x},\tilde{y})], \tag{14}$$

with $0 \leq C_1 \leq C_2$.

Indeed, consistency-based methods minimise an unsupervised objective that is a divergence between the model predictions and a modified version of the input (data augmentation) or a perturbation of the model. Using the fact that all norms are equivalent in a finite-dimensional space such as the space of the labels, we have the equivalence between a consistency-based $H$ and an expectation of $L$.

**VAT** The virtual adversarial training method proposed by (Miyato et al., 2018) generates the most impactful perturbation $r_{adv}$ to add to $x$. The objective is to train a model robust to input perturbations. This method is closely related to adversarial training introduced by Goodfellow et al. (2014).

$$H(\theta;x) = \mathbf{Div}(f_{\hat{\theta}}(x,.), f_\theta(x + r_{adv},.))$$

where the **Div** is a non-negative function that measures the divergence between two distributions, the cross-entropy or the KL divergence for instance. If the divergence function is the cross-entropy, it is straightforward to write the unlabelled objective as Equation 3. If the objective function is the KL divergence, we can write the objective as

$$H(\theta;x) = \mathbb{E}_{\pi_x(\tilde{x}+r,\tilde{y})}[L(\theta;\tilde{x},\tilde{y})] - \mathbb{E}_{\pi_x(\tilde{x},\tilde{y})}[L(\hat{\theta};\tilde{x},\tilde{y})] \tag{15}$$

with $\pi_x(\tilde{x},\tilde{y}) = \delta_x(\tilde{x})p_{\hat{\theta}}(y|x)$. Therefore, variation of $H$ with respect to $\theta$ are the same as $\mathbb{E}_{\pi_x(\tilde{x}+r,\tilde{y})}[L(\theta;\tilde{x},\tilde{y})]$. VAT is also a method between consistency-based and entropy-based methods as long as we use the KL-divergence or the cross-entropy as the measure of divergence.

**Mean-Teacher** A different form of pseudo-labelling is the Mean-Teacher approach proposed by (Tarvainen & Valpola, 2017) where pseudo-labels are generated by a teacher model for a student model. The parameters of the student model are updated, while the teacher's are a moving average of the student's parameters from the previous training steps. The idea is to have a more stable pseudo-labelling using the teacher than in the classic Pseudo-label. Final predictions are made by the student model. A generic form of the unsupervised part of the risk estimator is then

$$H(\theta;x) = \sum_y (p_\theta(y|x) - p_{\hat{\theta}}(y|x))^2,$$

where $\hat{\theta}$ are the fixed parameters of the teacher.

**$\Pi$-Model** The $\Pi$-Models are intrinsically stochastic models (for example a model with dropout) encouraged to make consistent predictions through several passes of the same $x$ in the model. The SSL loss is using the stochastic behaviour of the model where the model $f_\theta$ and penalises different predictions for the same $x$ (Sajjadi et al., 2016). Let's note $f_\theta(x,.)_1$ and $f_\theta(x,.)_2$ two passes of $x$ through the model $f_\theta$. A generic form of the unsupervised part of the risk estimator is then

$$H(\theta;x) = \mathbf{Div}(f_\theta(x,.)_1, f_{\hat{\theta}}(x,.)_2), \tag{16}$$

where **Div** is a measure of divergence between two distributions (often the Kullback-Leibler divergence).

**Temporal ensembling**   Temporal ensembling (Laine & Aila, 2017) is a form of $\Pi$-Model where we compare the current prediction of the model on the input $x$ with an accumulation of the previous passes through the model. Then, the training is faster as the network is evaluated only once per input on each epoch and the perturbation is expected to be less noisy than for $\Pi$-models.

**ICT**   Interpolation consistency training (Verma et al., 2019) is an SSL method based on the mixup operation (Zhang et al., 2017). The model trained is then consistent to predictions at interpolations. The unsupervised term of the objective is then computed on two terms:

$$H(\theta; x_1, x_2) = \mathbf{Div}\left(f_\theta(\alpha x_1 + (1 - \alpha)x_2, .), \alpha f_{\hat{\theta}}(x_1, .) + (1 - \alpha)f_{\hat{\theta}}(x_2, .)\right), \qquad (17)$$

with $\alpha$ drawn with from a distribution $\mathcal{B}(a, a)$. With the exact same transformation, we will be able to show that this objective is equivalent to a form of expectation of $L$.

## C  On the semi-supervised bias

We provide in this appendix a further explanation of the risk induced by the SSL bias as introduced in Section 2.3.

Presented methods minimise a biased version of the risk under the MCAR assumption and therefore classical learning theory does not apply anymore,

$$\mathbb{E}[\hat{\mathcal{R}}_{SSL}(\theta)] = \mathbb{E}[L(\theta; x, y)] + \lambda \mathbb{E}[H(\theta; x, y)] \neq \mathcal{R}(\theta). \tag{18}$$

Learning over a biased estimate of the risk is not necessarily unsafe but it is difficult to provide theoretical guarantees on such methods even if some works try to do so with strong assumptions on the data distribution (Mey & Loog 2019, Section 4 and 5, Zhang et al. 2021b). Previous works proposed generalisation error bounds of SSL methods under strong assumptions on the data distribution or the true model. We refer to the survey by Mey & Loog (2019). More recently, Wei et al. (2021) proves an upper bound for training deep models with the pseudo-label method under strong assumption. Under soft assumptions, Aminian et al. (2022) provides an error bound showing that the choice of $H$ is crucial to provide good performances.

Indeed, the unbiased nature of the risk estimate is crucial in the development of learning theory. This bias on the risk estimate may look like the one of a regularisation, such as the ridge regularisation. However, SSL and regularisation are intrinsically different for several reasons:

- Regularisers have a vanishing impact in the limit of infinite data whereas SSL usually do not in the proposed methods, see Equation 18. A solution would be to choose $\lambda$ with respect of the number of data points and make it vanish when $n$ goes to infinity. However, in most works, the choice of $\lambda$ is independent of the number of $n$ or $n_l$ (Oliver et al., 2018; Sohn et al., 2020).

- One of the main advantages of regularisation is to turn the learning problem into a "more convex" problem, see Shalev-Shwartz & Ben-David (2014, Chapter 13). Indeed, ridge regularisation will often turn a convex problem into a strongly-convex problem. However, SSL faces the danger to turn the learning problem as non-convex as previously noted by Sokolovska et al. (2008).

- The objective of a regulariser is to bias the risk towards optimum with smooth decision functions whereas entropy-based SSL will lead to sharp decision functions.

- Regularisation usually does not depend on the data whereas $H$ does in the SSL framework.

A entropy bias has been actually used by Pereyra et al. (2017) as a regulariser but as entropy *maximisation* which should has an effect that is the opposite of the SSL method introduced by Grandvalet & Bengio (2004), the entropy minimisation.

 # D   Proof that $\hat{\mathcal{R}}_{DeSSL}(\theta)$ is unbiased under MCAR

748 **Theorem D.1.** *Under the MCAR assumption, $\hat{\mathcal{R}}_{DeSSL}(\theta)$ is an unbiased estimator of $\mathcal{R}(\theta)$.*

749 As a consequence of the theorem, under the MCAR assumption, $\hat{\mathcal{R}}_{CC}(\theta)$ is also unbiased as a special
750 case of $\hat{\mathcal{R}}_{DeSSL}(\theta)$ for $\lambda = 0$

751 **Proof:**   We first recall that the DeSSL risk estimator $\hat{\mathcal{R}}_{DeSSL}(\theta)$ is defined for any $\lambda$ by

$$
\begin{aligned}
\hat{\mathcal{R}}_{DeSSL}(\theta) &= \frac{1}{n_l} \sum_{i=1}^{n_l} L(\theta; x_i, y_i) + \frac{\lambda}{n_u} \sum_{i=1}^{n_u} H(\theta; x_i) - \frac{\lambda}{n_l} \sum_{i=1}^{n_l} H(\theta; x_i) \\
&= \sum_{i=1}^{n} \left( \frac{r_i}{n_l} L(\theta; x_i, y_i) + \lambda \left( \frac{1 - r_i}{n_u} - \frac{r_i}{n_l} \right) H(\theta; x_i) \right).
\end{aligned}
\tag{19}
$$

752 By the law of total expectation:

$$
\mathbb{E}[\hat{\mathcal{R}}_{DeSSL}(\theta)] = \mathbb{E}_r \left[ \mathbb{E}_{x,y}[\hat{\mathcal{R}}_{DeSSL}(\theta)|r] \right].
$$

753 As far as we are under the MCAR assumption, the data $(x, y)$ and the missingness variable $r$ are
754 independent thus, $\mathbb{E}_r \left[ \mathbb{E}_{x,y}[\hat{\mathcal{R}}_{DeSSL}(\theta)|r] \right] = \mathbb{E}_r \left[ \mathbb{E}_{x,y}[\hat{\mathcal{R}}_{DeSSL}(\theta)] \right]$.

755 We focus on $\mathbb{E}_{x,y}[\hat{\mathcal{R}}_{DeSSL}(\theta)]$.   First, we replace $\hat{\mathcal{R}}_{DeSSL}(\theta)$ by its definition and then use the
756 linearity of the expectation. Then,

$$
\begin{aligned}
\mathbb{E}_{x,y}[\hat{\mathcal{R}}_{DeSSL}(\theta)] &= \mathbb{E} \left[ \frac{1}{n_l} \sum_{i=1}^{n_l} L(\theta; x_i, y_i) + \frac{\lambda}{n_u} \sum_{i=1}^{n_u} H(\theta; x_i) - \frac{\lambda}{n_l} \sum_{i=1}^{n_l} H(\theta; x_i) \right] && \text{by definition} \\
&= \frac{1}{n_l} \sum_{i=1}^{n_l} \mathbb{E} \left[ L(\theta; x_i, y_i) \right] + \frac{\lambda}{n_u} \sum_{i=1}^{n_u} \mathbb{E} \left[ H(\theta; x_i) \right] - \frac{\lambda}{n_l} \sum_{i=1}^{n_l} \mathbb{E} \left[ H(\theta; x_i) \right] && \text{by linearity}
\end{aligned}
$$

757 The couples $(x_i, y_i)$ are i.i.d. samples following the same distribution. Then, we have

$$
\begin{aligned}
\mathbb{E}_{x,y}[\hat{\mathcal{R}}_{DeSSL}(\theta)] &= \frac{1}{n_l} \sum_{i=1}^{n_l} \mathbb{E} \left[ L(\theta; x, y) \right] + \frac{\lambda}{n_u} \sum_{i=1}^{n_u} \mathbb{E} \left[ H(\theta; x) \right] - \frac{\lambda}{n_l} \sum_{i=1}^{n_l} \mathbb{E} \left[ H(\theta; x) \right] && \text{i.i.d samples} \\
&= \mathbb{E} \left[ L(\theta; x, y) \right] \\
&= \mathcal{R}(\theta).
\end{aligned}
$$

758 Finally, we have the results that , $\hat{\mathcal{R}}_{DeSSL}(\theta)$ is unbiased as $\mathcal{R}(\theta)$ is a constant,

$$
\mathbb{E}[\hat{\mathcal{R}}_{DeSSL}(\theta)] = \mathbb{E} \left[ \mathbb{E}_{x,y}[\hat{\mathcal{R}}_{DeSSL}(\theta)]|r \right] = \mathbb{E}_r \left[ \mathcal{R}(\theta) \right] = \mathcal{R}(\theta).
\tag{20}
$$

 # E  Proof and comments about Theorem 3.1

**Theorem 3.1**    *The function $\lambda \mapsto \mathbb{V}(\hat{\mathcal{R}}_{DeSSL}(\theta)|r)$ reaches its minimum for:*

$$\lambda_{opt} = \frac{n_u}{n} \frac{\mathrm{Cov}(L(\theta; x, y), H(\theta; x))}{\mathbb{V}(H(\theta; x))} \tag{21}$$

*and*

$$\mathbb{V}(\hat{\mathcal{R}}_{DeSSL}(\theta)|r)|_{\lambda_{opt}} = \left(1 - \frac{n_u}{n}\rho_{L,H}^2\right) \mathbb{V}(\hat{\mathcal{R}}_{CC}(\theta))$$
$$\leq \mathbb{V}(\hat{\mathcal{R}}_{CC}(\theta)), \tag{22}$$

*where $\rho_{L,H} = \mathrm{Corr}(L(\theta; x, y), H(\theta; x))$.*

**Proof:**    For any $\lambda \in \mathbb{R}$, we want to compute the variance:

$$\mathbb{V}(\hat{\mathcal{R}}_{DeSSL}(\theta)|r).$$

Under the MCAR assumption, $x$ and $y$ are both jointly independent of $r$. Also, the couples $(x_i, y_i, r_i)$ are independent. Therefore, we have

$$\mathbb{V}(\hat{\mathcal{R}}_{DeSSL}(\theta)|r) = \sum_{i=1}^{n} \mathbb{V}_{(x_i,y_i)\sim p(x,y|r)}\left(\frac{r_i}{n_l}L(\theta, x_i, y_i) + \lambda\left(\frac{1-r_i}{n_u} - \frac{r_i}{n_l}\right)H(\theta, x_i)\right) \quad \text{i.i.d samples}$$

$$= \sum_{i=1}^{n} \mathbb{V}_{(x_i,y_i)\sim p(x,y)}\left(\frac{r_i}{n_l}L(\theta, x_i, y_i) + \lambda\left(\frac{1-r_i}{n_u} - \frac{r_i}{n_l}\right)H(\theta, x_i)\right) \quad (x, y) \text{ and } r \text{ independent}$$

Using the fact that the couples $(x_i, y_i)$ are i.i.d. samples following the same distribution, we have

$$\mathbb{V}(\hat{\mathcal{R}}_{DeSSL}(\theta)|r) = \sum_{i=1}^{n} \mathbb{V}_{(x,y)\sim p(x,y)}\left(\frac{r_i}{n_l}L(\theta, x, y) + \lambda\left(\frac{1-r_i}{n_u} - \frac{r_i}{n_l}\right)H(\theta, x)\right)$$

$$= \sum_{i=1}^{n} \frac{r_i^2}{n_l^2}\mathbb{V}(L(\theta, x, y)) + \lambda^2\left(\frac{1-r_i}{n_u} - \frac{r_i}{n_l}\right)^2 \mathbb{V}(H(\theta, x)) \quad \text{using covariance}$$

$$+ 2\lambda\frac{r_i}{n_l}\left(\frac{1-r_i}{n_u} - \frac{r_i}{n_l}\right)\mathrm{Cov}(L(\theta, x, y), H(\theta, x))$$

Now, we remark that the variable $r$ is binary and therefore $r^2 = r$, $(1-r)^2 = 1 - r$ and $r(1-r) = 0$. Using that and simplifying, we have

$$\mathbb{V}(\hat{\mathcal{R}}_{DeSSL}(\theta)|r) = \sum_{i=1}^{n} \frac{r_i}{n_l^2}\mathbb{V}(L(\theta, x, y)) + \lambda^2\frac{(1-r_i)n_l^2 + r_i n_u^2}{n_l^2 n_u^2}\mathbb{V}(H(\theta, x))$$

$$- 2\lambda\frac{r_i}{n_l^2}\mathrm{Cov}(L(\theta, x, y), H(\theta, x))$$

Finally, by summing and simplifying the expression (note that $n_l + n_u = n$), we compute the expression variance,

$$\mathbb{V}(\hat{\mathcal{R}}_{DeSSL}(\theta)|r) = \frac{1}{n_l}\mathbb{V}(L(\theta, x, y)) + \lambda^2\frac{n}{n_l n_u}\mathbb{V}(H(\theta, x)) - \frac{2\lambda}{n_l}\mathrm{Cov}(L(\theta, x, y), H(\theta, x))$$

So $\mathbb{V}(\hat{\mathcal{R}}_{DeSSL}(\theta)|r)$ is a quadratic function in $\lambda$ and reaches its minimum for $\lambda_{opt}$ such that:

$$\lambda_{opt} = \frac{n_u}{n}\frac{\mathrm{Cov}(L(\theta, x, y), H(\theta, x))}{\mathbb{V}(H(\theta, x))}$$

And, at $\lambda_{opt}$, the variance of $\hat{\mathcal{R}}_{DeSSL}(\theta)|r)$ becomes

$$\mathbb{V}(\hat{\mathcal{R}}_{DeSSL}(\theta)|r) = \frac{1}{n_l}\mathbb{V}(L(\theta,x,y))\left(1 - \frac{n_u}{n}\frac{\text{Cov}(L(\theta,x,y),H(\theta,x))^2}{\mathbb{V}(H(\theta,x))\mathbb{V}(L(\theta;x,y))}\right)$$

$$= \frac{1}{n_l}\mathbb{V}(L(\theta,x,y))\left(1 - \frac{n_u}{n}\text{Corr}(L(\theta,x,y),H(\theta,x))^2\right)$$

$$= \left(1 - \frac{n_u}{n}\rho_{L,H}^2\right)\frac{1}{n_l}\mathbb{V}(L(\theta,x,y))$$

*Remark* E.1. If $H$ is perfectly correlated with $L$ ($\rho_{L,H} = 1$), then the variance of the DeSSL estimator is equal to the variance of the estimator with no missing labels.

*Remark* E.2. **Is it possible to estimate** $\lambda_{opt}$ **in practice ?** The data distribution $p(x,y)$ being unknown, the computation of $\lambda_{opt}$ is not possible directly. Therefore, we need to use an estimator of the covariance $\text{Cov}(L(\theta;x,y),H(\theta;x))$ and the variance $\mathbb{V}(H(\theta;x))$ (See Equation 23). Also, we have to be careful not to introduce a new bias with the computation of $\lambda_{opt}$, indeed, if we compute it using the training set, $\lambda_{opt}$ becomes dependent of $x$ and $y$ and therefore $\hat{\mathcal{R}}_{DeSSL}(\theta)|r)$ becomes biased. A solution would be to use a validation dataset for its computation. Another approach is to compute it using the splitting method (Avramidis & Wilson, 1993). Moreover, the computation of $\lambda_{opt}$ is tiresome and time-consuming in practice as it has to be updated for every different value of $\theta$, so at each gradient step.

$$\hat{\lambda}_{opt} = \frac{\frac{1}{n_l}\sum_{i=1}^{n_l}(L(\theta;x_i,y_i) - \bar{L}(\theta))(H(\theta;x_i) - \bar{H}(\theta))}{\frac{1}{n}\sum_{i=1}^{n}(H(\theta;x_i) - \bar{H}(\theta))^2} \tag{23}$$

where $\bar{H}(\theta) = \frac{1}{n}\sum_{i=1}^{n}H(\theta;x_i)$ and $\bar{L}(\theta) = \frac{1}{n_l}\sum_{i=1}^{n_l}L(\theta;x_i,y_i)$

*Remark* E.3. **About the sign of** $\lambda$ As explained in the article, the theorem still has a *quantitative* merit when it comes to choosing $\lambda$, by telling that the sign of $\lambda$ is positive when $H$ and $L$ are positively correlated which will generally be the case with the examples mentioned in the article. For instance, concerning the entropy minimisation technique, the following proposition proves that the log-likelihood is negatively correlated with its entropy and therefore it justifies the choice of $\lambda > 0$ in the entropy minimisation.

**Proposition E.4.** *The log-likelihood of the true distribution* $\log p(y|x)$ *is negatively correlated with its entropy* $\mathbb{H}_{\tilde{y}}(p(\tilde{y}|x)) = -\mathbb{E}_{\tilde{y}\sim p(.|x)}[\log p(\tilde{y}|x)]$.

$$\text{Cov}(\log p(y|x), \mathbb{H}_{\tilde{y}}(p(\tilde{y}|x))) < 0 \tag{24}$$

*Proof.*
$$\text{Cov}(\log p(y|x), \mathbb{H}_{\tilde{y}}(p(\tilde{y}|x))) = \mathbb{E}_{x,y}[\log p(y|x)\mathbb{H}_{\tilde{y}}(p(\tilde{y}|x))] - \mathbb{E}_{x,y}[\log p(y|x)]\mathbb{E}_x[\mathbb{H}_{\tilde{y}}(p(\tilde{y}|x))] \tag{25}$$

$$= -\mathbb{E}_{x,y}[\log p(y|x)\mathbb{E}_{\tilde{y}|x}[\log p(\tilde{y}|x)]] + \mathbb{E}_{x,y}[\log p(y|x)]\mathbb{E}_x[\mathbb{E}_{\tilde{y}|x}[\log p(\tilde{y}|x)]] \tag{26}$$

$$\tag{27}$$

By the law of total expectation, we have that $\mathbb{E}_x[\mathbb{E}_{\tilde{y}|x}[\log p(\tilde{y}|x)]] = \mathbb{E}_{x,\tilde{y}}[\log p(\tilde{y}|x)]$, then

$$\text{Cov}(\log p(y|x), \mathbb{H}_{\tilde{y}}(p(\tilde{y}|x)) = -\mathbb{E}_{x,y}[\log p(y|x)\mathbb{E}_{\tilde{y}|x}[\log p(\tilde{y}|x)]] + \mathbb{E}_{x,y}[\log p(y|x)]^2 \tag{28}$$

$$= \mathbb{E}_{x,y}[\log p(y|x)]^2 - \mathbb{E}_{x,y}[\log p(y|x)\mathbb{E}_{\tilde{y}|x}[\log p(\tilde{y}|x)]] \tag{29}$$

$$\tag{30}$$

On the other hand, also with the law of total expectation, $\mathbb{E}_{x,y}[\log p(y|x)\mathbb{E}_{\tilde{y}|x}[\log p(\tilde{y}|x)]] = \mathbb{E}_x[\mathbb{E}_{y|x}[\log p(y|x)]\mathbb{E}_{\tilde{y}|x}[\log p(\tilde{y}|x)]]$, so

$$\mathbb{E}_{x,y}[\log p(y|x)\mathbb{E}_{\tilde{y}|x}[\log p(\tilde{y}|x)]] = \mathbb{E}_x[\mathbb{E}_{y|x}[\log p(y|x)]^2]$$
$$\geq \mathbb{E}_x[\mathbb{E}_{y|x}[\log p(y|x)]]^2 \qquad \text{Jensen's inequality}$$
$$\geq \mathbb{E}_{x,y}[\log p(y|x)]^2 \qquad \text{total expectation law}$$

796    Finally, we have the results,

$$\text{Cov}(\log p(y|x), \mathbb{H}_{\tilde{y}}(p(\tilde{y}|x))) \leq \mathbb{E}_{x,y}[\log p(y|x)]^2 - \mathbb{E}_{x,y}[\log p(y|x)]^2$$
$$\leq 0$$

797    $\square$

798    *Remark* E.5. We can also see the Pseudo-label as a form of entropy. Indeed, modulo the confidence
799    selection on the predicted probability, the Pseudo-label objective is the inverse of the Rényi min-
800    entropy:

$$\mathbb{H}_{\infty}(x) = -\max_{y} \log p(y|x)$$

 # F   Why debiasing with the labelled dataset?

We remark that the debiasing can be performed with any subset of the training data, labelled and unlabelled. The choice of debiasing only with the labelled data can be explained both intuitively and computationally in regard to the Theorem 3.1. Intuitively, the debiasing term penalises the confidence on the labelled datapoints and then prevents the overfitting on the train dataset. As remarked in section 3.1, Pereyra et al. (2017) showed that penalising low entropy models acts as a strong regulariser in supervised settings. This comforts the idea of penalising low entropy on the labelled dataset, i.e. debiaising the entropy minimisation with the labelled dataset. Considering Pseudo-Label-based methods, the objective for the labelled data is to predict the correct labels with moderate confidence. This is also similar to the concept of plausibility inference described by Barndorff-Nielsen (1976).

In regard to Theorem 3.1, we show that the optimum choice of subset for debiaising is either only the labelled data or the whole dataset and both are equivalent.

We consider a subset $\mathcal{A}$ of the training set. We defined $a$ as follow:

$$a_i = \begin{cases} 1/|\mathcal{A}| & \text{if } x_i \in \mathcal{A} \\ 0 & \text{otherwise} \end{cases}.$$

The unbiased estimator is then:

$$\hat{\mathcal{R}}_{DeSSL,\mathcal{A}}(\theta) = \frac{1}{n_l}\sum_{i=1}^{n_l} L(\theta; x_i, y_i) + \frac{\lambda}{n_u}\sum_{i=1}^{n_u} H(\theta; x_i) - \lambda \sum_{i=1}^{n} a_i H(\theta; x_i). \tag{31}$$

We compute the variance of this quantity as in the proof of Theorem 3.1 and show that:

$$\mathbb{V}(\hat{\mathcal{R}}_{DeSSL,\mathcal{A}}(\theta)|r) = \sum_{i=1}^{n} \frac{r_i}{n_l^2}\mathbb{V}(L(\theta,x,y)) + \lambda^2\left(\frac{1-r_i}{n_u} - a_i\right)^2 \mathbb{V}(H(\theta,x)) - 2\lambda\frac{r_i a_i}{n_l}\text{Cov}(L(\theta,x,y),H(\theta,x)) \tag{32}$$

Suppose that no labelled datapoints are in $\mathcal{A}$. Then, the last term of the variance is null. Hence, having no labelled datapoints in $\mathcal{A}$ leads to a variance increase. We also remark that debiasing with the entire dataset is equivalent that debiasing with the labelled datapoints. Indeed

$$\begin{aligned}\hat{\mathcal{R}}_{DeSSL}(\theta) &= \frac{1}{n_l}\sum_{i=1}^{n_l} L(\theta; x_i, y_i) + \frac{\lambda}{n_u}\sum_{i=1}^{n_u} H(\theta; x_i) - \frac{1}{n}\sum_{i=1}^{n} H(\theta; x_i) \\ &= \frac{1}{n_l}\sum_{i=1}^{n_l} L(\theta; x_i, y_i) + \frac{\lambda}{n_u}\sum_{i=1}^{n_u} H(\theta; x_i) - \frac{\lambda}{n}\sum_{i=1}^{n_l} H(\theta; x_i) - \frac{\lambda}{n}\sum_{i=1}^{n_u} H(\theta; x_i) \\ &= \frac{1}{n_l}\sum_{i=1}^{n_l} L(\theta; x_i, y_i) + \frac{\lambda n_l}{n n_u}\sum_{i=1}^{n_u} H(\theta; x_i) - \frac{\lambda n_l}{n n_l}\sum_{i=1}^{n_l} H(\theta; x_i),\end{aligned}$$

which is equivalent to debiasing with only the labelled dataset by replacing $\lambda$ by $\lambda\frac{n_l}{n}$.

At this point we can still sample a random subset composed of $l$ labelled datapoints and $u$ unlabelled datapoints. Therefore $a_i = 1/(l+u)\mathbf{1}\{x_i \in \mathcal{A}\}$, we show in the following that the optimum choice of the couple $(l,u)$ are $(n_l, 0)$ and $(n_l, n_U)$, so only the labelled or the whole dataset.

We sample $l$ labelled and $u$ unlabelled datapoints to debiased the estimator, by simplifying the term in the sum of Equation 32 as follow:

$$\left(\frac{1-r_i}{n_u} - a_i\right)^2 = \begin{cases} \left(\frac{1}{n_u} - \frac{1}{l+u}\right)^2 & \text{if } x_i \in \mathcal{A} \text{ and } r_i = 0 \\ \frac{1}{(l+u)^2} & \text{if } x_i \in \mathcal{A} \text{ and } r_i = 1 \\ \frac{1}{n_u^2} & \text{if } x_i \notin \mathcal{A} \text{ and } r_i = 0 \\ 0 & \text{if } x_i \notin \mathcal{A} \text{ and } r_i = 1 \end{cases}.$$

823    Then, by summing the term and simplifying, we get:

$$\mathbb{V}(\hat{\mathcal{R}}_{DeSSL}(\theta)|r) = \frac{1}{n_l}\mathbb{V}(L(\theta,x,y)) + \lambda^2 \left[ u\left(\frac{1}{n_u} - \frac{1}{l+u}\right) + \frac{l}{(l+u)^2} + \frac{n_u - u}{n_u^2} \right] \mathbb{V}(H(\theta,x))$$

$$- 2\lambda \frac{l}{n_l(l+u)}\text{Cov}(L(\theta,x,y), H(\theta,x))$$

$$= \frac{1}{n_l}\mathbb{V}(L(\theta,x,y)) + \lambda^2 \frac{n_l}{n_u}\frac{n_u - u + l}{l+u}\mathbb{V}(H(\theta,x)) - 2\lambda\frac{l}{n_l(l+u)}\text{Cov}(L(\theta,x,y), H(\theta,x))$$

We want to minimise $\mathbb{V}(\hat{\mathcal{R}}_{DeSSL}(\theta)|r)$ with respect to $(\lambda, l, u)$. $\mathbb{V}(\hat{\mathcal{R}}_{DeSSL}(\theta)|r)$ reaches is minimum in $\lambda$ at

$$\lambda_{opt} = \frac{n_u}{n_l}\frac{l}{n_u - u + l}\frac{\text{Cov}(L(\theta,x,y), H(\theta,x))}{\mathbb{V}(H(\theta,x))}.$$

824    Then,

$$\mathbb{V}(\hat{\mathcal{R}}_{DeSSL}(\theta)|r) = \frac{1}{n_l}\mathbb{V}(L(\theta,x,y)) - \frac{n_u}{n_l}\frac{l^2}{(n_u - u + l)(l+u)}\frac{\text{Cov}(L(\theta,x,y), H(\theta,x))^2}{\mathbb{V}(H(\theta,x))}.$$

825    We now want to minimise with respect to $0 \leq u \leq n_u$ and $1 \leq l \leq n_l$. We can easily show that the
826    $(n_u - u + l)(l + u)$ reaches its mininum for $u = 0$ or $u = n_u$ and for both value:

$$\mathbb{V}(\hat{\mathcal{R}}_{DeSSL}(\theta)|r) = \frac{1}{n_l}\mathbb{V}(L(\theta,x,y)) - \frac{n_u}{n_l}\frac{l}{n_u + l}\frac{\text{Cov}(L(\theta,x,y), H(\theta,x))^2}{\mathbb{V}(H(\theta,x))}.$$

827    Then $l/(n_u + l)$ is a increasing function, then reaches its maximum a $l = n_l$. So finally, the optimal
828    choices for the couple $(n_l, 0)$ and $(n_l, n_u)$. We showed that these couples are equivalent.

 **G   Proof of Theorem 3.2**

830 **Theorem 3.2**   *If $\mathcal{S}(p_\theta, (x, y)) = -L(\theta; x, y)$ is a proper scoring rule, then*

$$\mathcal{S}'(p_\theta, (x, y, r)) = -(\frac{rn}{n_l}L(\theta; x, y) + \lambda n(\frac{1-r}{n_u} - \frac{r}{n_l})H(\theta; x)) \tag{33}$$

831 *is also a proper scoring rule.*

*Proof.* The scoring rule considered in our SSL framework is:

$$\mathcal{S}'(p_\theta, (x, y, r)) = -\left(\frac{rn}{n_l}L(\theta; x, y) + \lambda n(\frac{1-r}{n_u} - \frac{r}{n_l})H(\theta; x)\right).$$

The proper scoring rule of the fully supervised problem is

$$\mathcal{S}(p_\theta, (x, y, r)) = -L(\theta; x, y).$$

832 Let p be the true distribution of the data $(x, y, r)$. Under MCAR, $r$ is independent of $x$ and $y$, then
833 $p(x, y, r) = p(r)p(x, y)$.

$$\mathcal{S}'(p_\theta, p) = \int p(x, y, r)\mathcal{S}'(p_\theta, (x, y, r))\, dx\, dy\, dr$$

$$= \int p(x, y)p(r)\mathcal{S}'(p_\theta, (x, y, r))\, dx\, dy\, dr \qquad \text{by independence}$$

$$= -\int p(x, y)p(r)\frac{rn}{n_l}L(\theta; x, y) + \lambda n(\frac{1-r}{n_u} - \frac{r}{n_l})H(\theta; x)\, dx\, dy\, dr$$

$$= -\int_{x,y} p(x, y)\underbrace{\left(\int_r p(r)\frac{rn}{n_l}\, dr\right)}_{=1} L(\theta; x, y)\, dx\, dy$$

$$\qquad - \lambda n\int_{x,y} p(x, y)\underbrace{\left(\int_r p(r)\left(\frac{1-r}{n_u} - \frac{r}{n_l}\right)\right)\, dr\right)}_{=0} H(\theta; x)\, dx\, dy$$

$$= -\int_{x,y} p(x, y)L(\theta; x, y)\, dx\, dy$$

$$= \mathcal{S}(p_\theta, p)$$

834 Therefore, if $\mathcal{S}(p_\theta, (x, y)) = -L(\theta; x, y)$ is a proper scoring rule, then
835 $mathcalS'(p_\theta, (x, y, r)) = -(\frac{rn}{n_l}L(\theta; x, y) + \lambda n(\frac{1-r}{n_u} - \frac{r}{n_l})H(\theta; x))$ is also a proper scoring rule.

836 $\qquad\qquad\qquad\qquad\qquad\qquad\qquad\qquad\qquad\qquad\qquad\qquad\qquad\qquad\qquad\qquad\qquad\qquad\qquad\qquad\qquad\square$

 # H   Proof of Theorem 3.5

838 Assumption 3.3: the minimum $\theta^*$ of $\mathcal{R}$ is well-separated.

$$\inf_{\theta:d(\theta^*,\theta)\geq\epsilon} \mathcal{R}(\theta) > \mathcal{R}(\theta^*) \tag{34}$$

839 Assumption 3.4: uniform weak law of large numbers holds for a function $L$ if:

$$\sup_{\theta\in\Theta} \left| \frac{1}{n}\sum_{i=1}^{n} L(\theta, x_i, y_i) - \mathbb{E}[L(\theta, x, y)] \right| \xrightarrow{p} 0 \tag{35}$$

840 **Theorem 3.5.**   *Under assumption A and assumption B for both $L$ and $H$, $\hat{\theta} = \arg\min \hat{\mathcal{R}}_{DeSSL}$ is*
841 *asymptotically consistent with respect to $n$.*

842 This result is a direct application of Theorem 5.7 from van der Vaart (2000, Chapter 5) that states
843 that under assumption A and B for $L$, $\hat{\theta} = \arg\min \hat{\mathcal{R}}$ is asymptotically consistent with respect to $n$.
844 Assumption A remains unchanged as we have M-estimators of the same $\mathcal{R}$. We now aim to prove that
845 under assumption B for both $L$ and $H$, we have the assumption B on $\theta \longrightarrow \frac{rn}{n_l}L(\theta; x, y) + \lambda(1 -$
846 $\frac{rn}{n_l})H(\theta; x)$.

847 **Lemma H.1.**   *If the uniform law of large number holds for both $L$ and $H$, then it holds for $\theta \longrightarrow$*
848 $\frac{rn}{n_l}L(\theta; x, y) + \lambda(1 - \frac{rn}{n_l})H(\theta; x)$.

849 *Proof.*  Suppose assumption B for $L$, then the same result holds if we replace $n$ with $n_l$ as $n$ and $n_l$
850 are coupled by the law of $r$. Indeed, when $n$ grows to infinity, $n_l$ too and inversely. Therefore,

$$\sup_{\theta\in\Theta} \left| \frac{1}{n_l}\sum_{i=1}^{n_l} L(\theta; x_i, y_i) - \mathbb{E}[L(\theta; x, y)] \right| \xrightarrow[n]{p} 0$$

851 Now, suppose we have assumption B for $H$, then we can make the same remark than for $L$. Now, we
852 have to show that:

$$\sup_{\theta\in\Theta} \left| \frac{1}{n}\sum_{i=1}^{n} \frac{rn}{n_l}L(\theta; x, y) + \lambda n \left( \frac{1-r}{n_u} - \frac{r}{n_l} \right) H(\theta; x) - \mathbb{E}[L(\theta; x, y)] \right| \xrightarrow[n]{p} 0$$

853 We first split the absolute value and the sup operator as

$$\sup_{\theta\in\Theta} \left| \frac{1}{n}\sum_{i=1}^{n} \frac{rn}{n_l}L(\theta; x, y) + \lambda n \left( \frac{1-r}{n_u} - \frac{r}{n_l} \right) H(\theta; x) - \mathbb{E}[L(\theta; x, y)] \right|$$

$$\leq \sup_{\theta\in\Theta} \left| \frac{1}{n_l}\sum_{i=1}^{n} \frac{rn}{n_l}L(\theta; x, y) - \mathbb{E}[L(\theta; x, y)] \right| + \left| \frac{1}{n}\sum_{i=1}^{n} \lambda n \left( \frac{1-r}{n_u} - \frac{r}{n_l} \right) H(\theta; x) \right|$$

$$\leq \underbrace{\sup_{\theta\in\Theta} \left| \frac{1}{n_l}\sum_{i=1}^{n_l} L(\theta; x, y) - \mathbb{E}[L(\theta; x, y)] \right|}_{\xrightarrow[n]{p} 0} + \sup_{\theta\in\Theta} \left| \frac{1}{n}\sum_{i=1}^{n} \lambda n \left( \frac{1-r}{n_u} - \frac{r}{n_l} \right) H(\theta; x) \right|.$$

854 So we now have to prove that the second term is also converging to $0$ in probability. Again by splitting
855 the absolute value and the sup, we have

$$\sup_{\theta\in\Theta} \left| \frac{1}{n}\sum_{i=1}^{n} \lambda n \left( \frac{1-r}{n_u} - \frac{r}{n_l} \right) H(\theta; x) \right| = \sup_{\theta\in\Theta} \left| \frac{\lambda}{n}\sum_{i=1}^{n} \frac{(1-r)n}{n_u} H(\theta; x) - \frac{\lambda}{n}\sum_{i=1}^{n} \frac{rn}{n_l} H(\theta; x) \right|$$

Then we have that,

$$\sup_{\theta \in \Theta} \left| \frac{\lambda}{n_u} \sum_{i=1}^{n} (1-r)H(\theta;x) - \frac{\lambda}{n_l} \sum_{i=1}^{n} rH(\theta;x) \right|$$

$$= \sup_{\theta \in \Theta} \left| \frac{\lambda}{n_u} \sum_{i=1}^{n} (1-r)H(\theta;x) - \mathbb{E}[H(\theta;x,y)] - \left( \frac{\lambda}{n_l} \sum_{i=1}^{n} rH(\theta;x) - \mathbb{E}[H(\theta;x,y)] \right) \right|$$

$$= \sup_{\theta \in \Theta} \left| \frac{\lambda}{n_u} \sum_{i=n_l+1}^{n_l+n_u} H(\theta;x) - \mathbb{E}[H(\theta;x,y)] - \left( \frac{\lambda}{n_l} \sum_{i=1}^{n_l} H(\theta;x) - \mathbb{E}[H(\theta;x,y)] \right) \right|$$

$$\leq \underbrace{\sup_{\theta \in \Theta} \left| \frac{\lambda}{n_u} \sum_{i=n-l+1}^{n_l+n_u} H(\theta;x) - \mathbb{E}[H(\theta;x,y)] \right|}_{\xrightarrow[n]{p} 0} + \underbrace{\sup_{\theta \in \Theta} \left| \left( \frac{\lambda}{n_l} \sum_{i=1}^{n_l} H(\theta;x) - \mathbb{E}[H(\theta;x,y)] \right) \right|}_{\xrightarrow[n]{p} 0}.$$

Thus,

$$\sup_{\theta \in \Theta} \left| \frac{1}{n} \sum_{i=1}^{n} \frac{rn}{n_l} L(\theta;x,y) + \lambda n \left( \frac{1-r}{n_u} 1 - \frac{r}{n_l} \right) H(\theta;x) - \mathbb{E}[L(\theta;x,y)] \right| \xrightarrow[n]{p} 0$$

And we now just have to apply the results of van der Vaart (2000, Theorem 5.7) to have the asymptotic consistent of $\hat{\theta} = \arg\min \hat{\mathcal{R}}_{DeSSL}$.

$\square$

*Remark* H.2. A sufficient condition on the function $H$ to verify assumption B, the uniform weak law of large numbers, is to be bounded (Newey & McFadden, 1994, Lemma 2.4). For instance, the entropy $H = - \sum_y p_\theta(y|x) \log(p_\theta(y|x))$ is bounded and therefore, the entropy minimisation is asymptotically consistent.

 # I   Asymptotic normality of DeSSL

866 In the following, we study a modified version of the objective to simplify the proof. Let us consider
867 the following DeSSL objective $L'(\theta; x, y, r) = \frac{r}{\pi} L(\theta; x, y) + \lambda \left( \frac{1-r}{1-\pi} - \frac{r}{\pi} \right) H(\theta; x)$ which has the
868 same properties than the original one (unbiasedness, variance reduction property, consistency and
869 benefit from generalisation error bounds). The idea is to replace $n_l$ with $\pi n$ to simplify the expression.
870 The value $n_l$ converges to $\pi n$ then the following Theorem should hold with the true DeSSL objective.

871 We define the cross-covariance matrice between random vectors $\nabla L(\theta; x, y)$ and $\nabla H(\theta; x)$ as
872 $K_\theta(i, j) = \mathbf{Cov}(\nabla L(\theta; x, y)_i, \nabla H(\theta; x)_j)$.

873 **Theorem I.1.** *Suppose $L$ and $H$ are smooth functions in $\mathcal{C}^2(\Theta, \mathbb{R})$. Assume $\mathcal{R}(\theta)$ admit a second-*
874 *order Taylor expansion at $\theta^*$ with a non-singular second order derivative $V_{\theta^*}$. Under the MCAR*
875 *assumption, we have that $\hat{\theta}_{DeSSL}$ is asymptotically normal with covariance:*

$$\Sigma_{DeSSL} = \frac{1}{\pi} V_{\theta^*}^{-1} \mathbb{E} \left[ \nabla L(\theta^*; x, y) \nabla L(\theta^*; x, y)^T \right] V_{\theta^*}^{-1}$$
$$+ \frac{\lambda^2}{\pi(1-\pi)} V_{\theta^*}^{-1} \mathbb{E} \left[ \nabla H(\theta^*; x, y) \nabla H(\theta^*; x, y)^T \right] V_{\theta^*}^{-1}$$
$$- \frac{\lambda}{\pi} V_{\theta^*}^{-1} K_{\theta^*} V_{\theta^*}^{-1}.$$

876 *As a consequence, we can minimise the trace of the covariance. Indeed, $\mathbf{Tr}(\Sigma_{DeSSL})$ reaches its*
877 *minimum at*

$$\lambda_{opt} = (1 - \pi) \frac{\mathbf{Tr}(V_{\theta^*}^{-1} K_{\theta^*} V_{\theta^*}^{-1})}{\mathbf{Tr}(V_{\theta^*}^{-1} \mathbb{E} \left[ \nabla H(\theta^*; x) \nabla H(\theta^*; x)^T \right] V_{\theta^*}^{-1})}, \tag{36}$$

878 *and at $\lambda_{opt}$ :*

$$\mathbf{Tr}(\Sigma_{DeSSL}) - \mathbf{Tr}(\Sigma_{CC}) = -\frac{1 - \pi}{\pi} \frac{\mathbf{Tr}(V_{\theta^*}^{-1} K_{\theta^*} V_{\theta^*}^{-1})^2}{\mathbf{Tr}(V_{\theta^*}^{-1} \mathbb{E} \left[ \nabla H(\theta^*; x) \nabla H(\theta^*; x)^T \right] V_{\theta^*}^{-1})} \leq 0. \tag{37}$$

879 The complete case is the special case of DeSSL with $\lambda = 0$. Then, the Theorem holds for the
880 complete case.

881 *Proof.* We define $L'(\theta; x, y, r) = \frac{r}{\pi} L(\theta; x, y) + \lambda \left( \frac{1-r}{1-\pi} - \frac{r}{\pi} \right) H(\theta; x)$ The assumptions of the
882 theorem are sufficient assumptions to apply Theorem 5.23 of Van der Vaart 1998 to the couple
883 $(\hat{\theta}_{DeSSL}, L')$. Hence, we obtain the following representation for representation $\hat{\theta}_{DeSSL}$:

$$\sqrt{n}(\hat{\theta}_{DeSSL} - \theta^*) = \frac{1}{\sqrt{n}} V_{\theta^*}^{-1} \sum_{i=1}^{n} \frac{r_i}{\pi} \nabla L(\theta^*; x_i, y_i) + \lambda \left( \frac{1-r_i}{1-\pi} - \frac{r_i}{\pi} \right) \nabla H(\theta^*; x_i) + o_p(1). \tag{38}$$

884

$$\sqrt{n}(\hat{\theta}_{DeSSL} - \theta^*) \xrightarrow{\mathcal{L}} \mathcal{N}(0, \Sigma_{DeSSL}),$$

885 The asymptotic normality follows with variance:

$$\Sigma_{DeSSL} = V_{\theta^*}^{-1} \mathbb{E} \left[ \nabla L'(\theta^*; x, y) \nabla L'(\theta^*; x, y)^T \right] V_{\theta^*}^{-1}.$$

886 Using the MCAR assumption, we simplify the expression of $\Sigma_{DeSSL}$:

$$\Sigma_{DeSSL} = V_{\theta^*}^{-1} \mathbb{E}\left[\nabla L'(\theta^*; x, y)\nabla L'(\theta^*; x, y)^T\right] V_{\theta^*}^{-1}$$

$$= \frac{1}{\pi^2} V_{\theta^*}^{-1} \mathbb{E}\left[r\nabla L(\theta^*; x, y)\nabla L(\theta^*; x, y)^T\right] V_{\theta^*}^{-1}$$

$$+ \lambda^2 V_{\theta^*}^{-1} \mathbb{E}\left[\left(\frac{1-r}{(1-\pi)^2} + \frac{r}{\pi^2}\right)\nabla H(\theta^*; x, y)\nabla H(\theta^*; x, y)^T\right] V_{\theta^*}^{-1}$$

$$- \frac{\lambda}{\pi^2} V_{\theta^*}^{-1} \mathbb{E}\left[r\nabla L(\theta^*; x, y)\nabla H(\theta^*; x, y)^T\right] V_{\theta^*}^{-1}$$

$$= \frac{1}{\pi} V_{\theta^*}^{-1} \mathbf{Cov}(L(\theta^*; x, y)) V_{\theta^*}^{-1} + \frac{\lambda^2}{\pi(1-\pi)} V_{\theta^*}^{-1} \mathbb{E}\left[\nabla H(\theta^*; x, y)\nabla H(\theta^*; x, y)^T\right] V_{\theta^*}^{-1} - \frac{\lambda}{\pi} V_{\theta^*}^{-1} K_{\theta^*} V_{\theta^*}^{-1}.$$

We remark that the complete case is the particular case of DeSSL with $\lambda = 0$. Then,

$$\Sigma_{DeSSL} = \Sigma_{CC} + \frac{\lambda^2}{\pi(1-\pi)} V_{\theta^*}^{-1} \mathbb{E}\left[\nabla H(\theta^*; x, y)\nabla H(\theta^*; x, y)^T\right] V_{\theta^*}^{-1}$$

$$- \frac{\lambda}{\pi} V_{\theta^*}^{-1} K_{\theta^*} V_{\theta^*}^{-1}.$$

The asymptotic relative efficiency of consequence, the asymptotic relative efficiency $\hat{\theta}_{DeSSL}$ compared to $\hat{\theta}_{CC}$ is defined as the quotient $\frac{\mathbf{Tr}(\Sigma_{DeSSL})}{\mathbf{Tr}(\Sigma_{CC})}$. This quotien can be minimised with respect to $\lambda$:

$$\lambda_{opt} = (1-\pi)\frac{\mathbf{Tr}(V_{\theta^*}^{-1} K_{\theta^*} V_{\theta^*}^{-1})}{\mathbf{Tr}(V_{\theta^*}^{-1}\mathbb{E}\left[\nabla H(\theta^*; x)\nabla H(\theta^*; x)^T\right] V_{\theta^*}^{-1})}, \tag{39}$$

and at $\lambda_{opt}$ :

$$\frac{\mathbf{Tr}(\Sigma_{DeSSL})}{\mathbf{Tr}(\Sigma_{CC})} = 1 - \frac{1-\pi}{\pi}\frac{\mathbf{Tr}(V_{\theta^*}^{-1} K_{\theta^*} V_{\theta^*}^{-1})^2}{\mathbf{Tr}(V_{\theta^*}^{-1}\mathbb{E}\left[\nabla H(\theta^*; x)\nabla H(\theta^*; x)^T\right] V_{\theta^*}^{-1})\mathbf{Tr}(\Sigma_{CC})} \leq 1. \tag{40}$$

$\square$

*Remark* I.2. **On the sign of $\lambda$.** It is easy to show that a sufficient condition to have $\lambda_{opt} > 0$ is to have $K_{\theta^*}$ positive semi-definite. Indeed, using that $V_{\theta^*}$ is positive definite and Proposition 6.1 of Serre (2010), we show that $\mathbf{Tr}(V_{\theta^*}^{-1} K_{\theta^*} V_{\theta^*}^{-1}) > 0$ and then $\lambda_{opt} > 0$.

*Remark* I.3. **Why minimising the trace of $\Sigma_{DeSSL}$?** Minimising the trace of $\Sigma_{DeSSL}$ leads to an estimator with a smaller asymptotic MSE, see Chen et al. (2020).

*Remark* I.4. **Fully supervised setting.** We also remark that our theorem matches the theorem for the supervised setting. Indeed, observing all the labelled corresponds to the case $\pi = 1$ and we obtain:

$$\Sigma_{DeSSL} = \Sigma_{CC} = \Sigma_{\text{Fully supervised}}.$$

 # J  Proof of Theorem 3.6

Our proof will be based on the following result from Shalev-Shwartz & Ben-David (2014, Theorem 26.5).

**Theorem J.1.** *Let $\mathcal{H}$ be a set of parameters, $z \sim \mathcal{D}$ a random variable living in a space $\mathcal{Z}$, $c > 0$, and $\ell : \mathcal{H} \times \mathcal{Z} \longrightarrow [-c, c]$. We denote*

$$L_{\mathcal{D}}(h) = \mathbb{E}_z[\ell(h, z)], \ and \ L_{\mathcal{S}}(h) = \frac{1}{m} \sum_{i=1}^{m} \ell(h, z_i), \tag{41}$$

*where $z_1, ..., z_m$ are i.i.d. samples from $\mathcal{D}$. For any $\delta > 0$, with probability at least $1 - \delta$, we have*

$$L_{\mathcal{D}}(h) \leq L_{\mathcal{S}}(h) + 2\mathbb{E}_{(\varepsilon_i)_{i \leq m}} \left[ \sup_{h \in \mathcal{H}} \left( \frac{1}{m} \sum_{i=1}^{m} \varepsilon_i \ell(h, z_i) \right) \right] + 4c \sqrt{\frac{2 \log(4/\delta)}{m}}, \tag{42}$$

*where $\varepsilon_1, ..., \varepsilon_m$ are i.i.d. Rademacher variables independent from $z_1, ..., z_m$.*

We can now restate and prove our generalisation bound.

**Theorem 3.6.** *We assume that both $L$ and $H$ are bounded and that the labels are MCAR. Then, there exists a constant $\kappa > 0$, that depends on $\lambda$, $L$, $H$, and the ratio of observed labels, such that, with probability at least $1 - \delta$, for all $\theta \in \Theta$,*

$$\mathcal{R}(\theta) \leq \hat{\mathcal{R}}_{DeSSL}(\theta) + 2R_n + \kappa \sqrt{\frac{\log(4/\delta)}{n}}, \tag{43}$$

*where $R_n$ is the Rademacher complexity*

$$R_n = \mathbb{E}_{(\varepsilon_i)_{i \leq n}} \left[ \sup_{\theta \in \Theta} \left( \frac{1}{n_l} \sum_{i=1}^{n_l} \varepsilon_i L(\theta; x_i, y_i) - \frac{\lambda}{n_l} \sum_{i=1}^{n_l} \varepsilon_i H(\theta; x_i) + \frac{\lambda}{n_u} \sum_{i=1}^{n_u} \varepsilon_i H(\theta; x_i) \right) \right], \tag{44}$$

*with $\varepsilon_1, ..., \varepsilon_m$ i.i.d. Rademacher variables independent from the data.*

*Proof.* We use Theorem J.1 with $z = (x, y, r)$, $\mathcal{H} = \Theta$, $m = n$, and

$$\ell(h, z) = \frac{nr_i}{n_l} L(\theta; x_i, y_i) + \lambda \left( \frac{n(1 - r_i)}{n_u} - \frac{nr_i}{n_l} \right) H(\theta; x_i). \tag{45}$$

The unbiasedness of our estimate under the MCAR assumption, proven in Appendix D, ensures that the condition of Equation (41) is satisfied with $L_{\mathcal{D}}(h) = \mathcal{R}(\theta)$ and $L_S(h) = \hat{\mathcal{R}}_{DeSSL}(\theta)$. Now, since $L$ and $H$ are bounded, there exists $M > 0$ such that $|L| < M$ and $|H| < M$. We can then bound $\ell$:

$$|\ell(h, z)| \leq \frac{n}{n_l} M + \lambda \max \left\{ \frac{n}{n_u}, \frac{n}{n_l} \right\} M = c. \tag{46}$$

Now that we have chosen a $c$ that bounds $\ell$, we can use Theorem J.1 and finally get Equation (43) with $\kappa = 4c\sqrt{2}$. $\qquad \square$

## K  DeSSL with $H$ applied on all available data

For consistency-based SSL methods it is common to use all the available data for the consistency term:

$$\hat{\mathcal{R}}_{SSL}(\theta) = \frac{1}{n_l} \sum_{i=1}^{n_l} L(\theta; x_i, y_i) + \frac{\lambda}{n} \sum_{i=1}^{n} H(\theta; x_i). \tag{47}$$

With the same idea, we debias the risk estimate with the labelled data:

$$\hat{\mathcal{R}}_{DeSSL}(\theta) = \frac{1}{n_l} \sum_{i=1}^{n_l} L(\theta; x_i, y_i) + \frac{\lambda}{n} \sum_{i=1}^{n} H(\theta; x_i) \\ - \frac{\lambda}{n_l} \sum_{i=1}^{n_l} H(\theta; x_i). \tag{48}$$

Under MCAR, this risk estimate is unbiased and the main theorem of the article hold with minor modifications. In Theorem 3.1, $\lambda_{opt}$ is slightly different and the expression of the variance at $\lambda_{opt}$ remains the same. The scoring rule in Theorem 3.2 is different but the theorem remains the same. Both Theorem 3.5 and 3.6 remain the same with very similar proofs.

**Theorem K.1.** *The function* $\lambda \mapsto \mathbb{V}(\hat{\mathcal{R}}_{DeSSL}(\theta))$ *reaches its minimum for:*

$$\lambda_{opt} = \frac{\text{Cov}(L(\theta; x, y), H(\theta; x))}{\mathbb{V}(H(\theta; x))} \tag{49}$$

*and*

$$\mathbb{V}(\hat{\mathcal{R}}_{DeSSL}(\theta))|_{\lambda_{opt}} = (1 - \frac{n_u}{n} \rho_{L,H}^2) \mathbb{V}(\hat{\mathcal{R}}_{CC}(\theta)) \\ \leq \mathbb{V}(\hat{\mathcal{R}}_{CC}(\theta)) \tag{50}$$

*where* $\rho_{L,H} = \text{Corr}(L(\theta; x, y), H(\theta; x))$.

When $H$ is applied on all labelled and unlabelled data, the scoring rule used in the learning process is then $\mathcal{S}'(p_\theta, (x, y, r)) = -(\frac{rn}{n_l} L(\theta; x, y) + \lambda(1 - \frac{rn}{n_l}) H(\theta; x))$ and we have $\mathcal{S}'$ is a proper scoring rule.

## L MNIST and MedMNIST

### L.1 MNIST

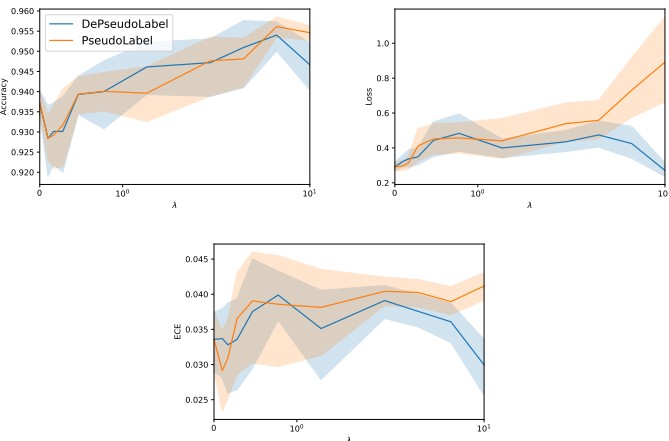

Figure 6: The influence of $\lambda$ on Pseudo-label and DePseudo-label for a Lenet trained on MNIST with $n_l = 1000$: (Left) Test accuracy; (Middle) Mean test cross-entropy; (Right) Mean test ECE, with 95% CI

### L.2 MNIST label noise

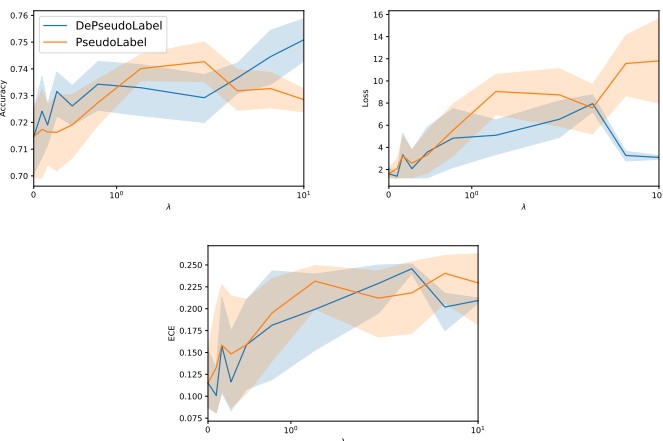

Figure 7: The influence of $\lambda$ on Pseudo-label and DePseudo-label for a Lenet trained on MNIST with label noise with $n_l = 1000$: (Left) Mean test accuracy; (Middle) Mean test cross-entropy; (Right) Test ECE, with 95% CI.

 **L.3   MedMNIST**

Table 3: Test AUC of Complete Case , PseudoLabel and DePseudoLabel on five datasets of MedM-NIST.

| DATASET | COMPLETE CASE | PSEUDOLABEL | DEPSEUDOLABEL |
|---|---|---|---|
| DERMA | $84.26 \pm 0.50$ | $82.64 \pm 1.19$ | $83.82 \pm 0.95$ |
| PNEUMONIA | $94.28 \pm 0.46$ | $94.34 \pm 0.91$ | $94.15 \pm 0.33$ |
| RETINA | $70.70 \pm 0.74$ | $70.12 \pm 1.01$ | $69.97 \pm 1.44$ |
| BREAST | $74.67 \pm 3.68$ | $74.86 \pm 3.18$ | $75.33 \pm 3.05$ |
| BLOOD | $97.83 \pm 0.23$ | $97.83 \pm 0.23$ | $97.72 \pm 0.15$ |

# M    PseudoLabel and DePseudoLabel on CIFAR: p-values

## M.1    CIFAR-10

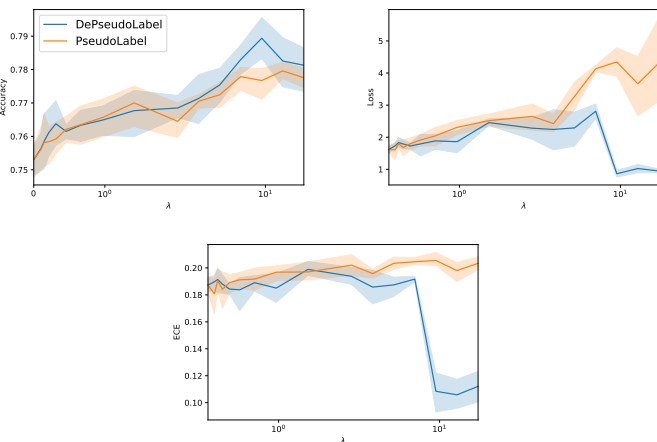

Figure 8: The influence of $\lambda$ on Pseudo-label and DePseudo-label on CIFAR-10 with nl= 4000: (Left) Mean test accuracy; (Middle) Mean test cross-entropy; (Right) Test ECE, with 95% CI.

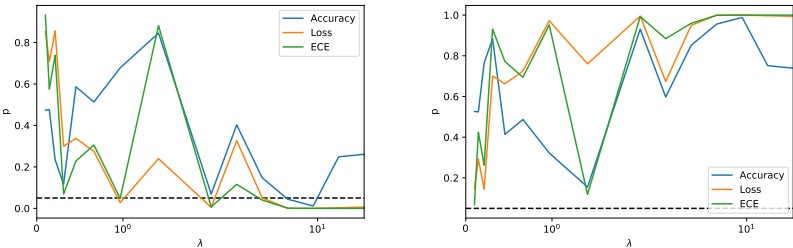

Figure 9: p-values of a paired student test between PseudoLabel and DePseudoLabel (Right) DePseudoLabel is better than PseudoLabel; (Left) DePseudoLabel is worst than PseudoLabel.

## M.2    Computation of $\lambda_{opt}$ on the test set.

As explained in the main text, the estimation of $\mathrm{Cov}(L(\theta; x, y), H(\theta; x))$ with few labels led to extremely unstable unsatisfactory results. However, we test the formula on CIFAR-10 and different methods to provide intuition on the order of $\lambda_{opt}$ and the range of the variance reduction regime (between 0 and $2\lambda_{opt}$). To do so, we estimate $\lambda_{opt}$ on the test set for CIFAR-10 by training a CNN13 using only $4,000$ labelled data on 200 epochs. The value of $\lambda_{opt}$ is 1.67, 31.16 and 0.66 for entropy minimisation, pseudo label and Fixmatch. Therefore, the reduced variance regime covers the intuitive choices of $\lambda$ in the SSL literature. Unfortunately, computing $\lambda_{opt}$ on the test set is not applicable in practice.

 **M.3 CIFAR-100**

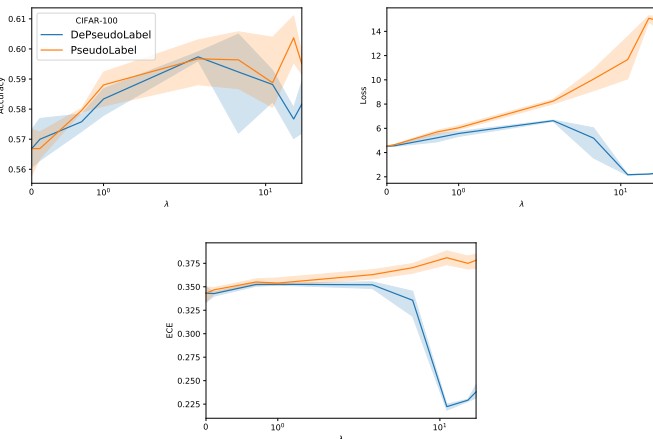

Figure 10: The influence of $\lambda$ on Pseudo-label and DePseudo-label on CIFAR-100 with nl= 4000: (Left) Mean test accuracy; (Middle) Mean test cross-entropy; (Right) Test ECE, with 95% CI.

# N Fixmatch (Sohn et al., 2020)

## N.1 Per class accuracy

In recent work, Zhu et al. (2022) exposed the disparate effect of SSL on different classes. Indeed, classes with a high complete case accuracy benefit more from SSL than classes with a low baseline accuracy. They introduced a metric called the benefit ratio ($\mathcal{BR}$) that quantifies the impact of SSL on a class $C$:

$$\mathcal{BR}(C) = \frac{acc_{SSL}(C) - acc_{CC}(C)}{acc_{S}(C) - acc_{CC}(C)}, \tag{51}$$

where $acc_{SSL}(C)$, $acc_{CC}(C)$ and $acc_{S}(C)$ are respectively the accuracy of the class with an SSL trained model, a complete-case model and a fully supervised model (a model that has access to all labels). Inspired by this work, we report the per class accuracy and the benefit ratio in Table N.1. We see that the "poor" classes such as bird, cat and dog tend to benefit from DeFixmatch much more than from Fixmatch. We compute $acc_{S}(C)$ using a pre-trained model with the same architecture[1]. Zhu et al. (2022) also promote the idea that a fair SSL algorithm should benefit different sub-classes equally, then having $\mathcal{BR}(C) = \mathcal{BR}(C')$ for all $C, C'$. While perfect equality seems unachievable in practice, we propose to look at the standard deviation of the $\mathcal{BR}$ through the different classes. While the standard deviation of Fixmatch is 0.12, the one of DeFixmatch is 0.06. Therefore, DeFixmatch improves the sub-populations' accuracies more equally.

Table 4: Mean accuracy per class and mean benefit ratio ($\mathcal{BR}$) on 5 folds for Fixmatch, DeFixmatch and the Complete Case. Bold: "poor" complete case accuracy classes.

|  | COMPLETE CASE | FIXMATCH | | DEFIXMATCH | |
| --- | --- | --- | --- | --- | --- |
|  | ACCURACY | ACCURACY | $\mathcal{BR}$ | ACCURACY | $\mathcal{BR}$ |
| AIRPLANE | 86.94 | 95.94 | 0.88 | 96.62 | 0.94 |
| AUTOMOBILE | 95.26 | 97.54 | 0.68 | 98.22 | 0.89 |
| **BIRD** | **80.46** | **90.80** | **0.68** | **92.64** | **0.80** |
| **CAT** | **70.08** | **82.50** | **0.56** | **87.16** | **0.78** |
| DEER | 88.88 | 95.86 | 0.78 | 97.26 | 0.94 |
| **DOG** | **79.66** | **87.16** | **0.53** | **90.98** | **0.81** |
| FROG | 93.12 | 97.84 | 0.80 | 98.62 | 0.94 |
| HORSE | 90.96 | 96.94 | 0.83 | 97.64 | 0.92 |
| SHIP | 94.12 | 97.26 | 0.67 | 98.06 | 0.84 |
| TRUCK | 93.18 | 96.82 | 0.84 | 97.20 | 0.93 |

## N.2 Fixmatch details

As first detailed in Appendix B, Fixmatch is a pseudo-label based method with data augmentation. Indeed, Fixmatch uses weak augmentations of $x$ (flip-and-shift) for the pseudo-labels selection and then minimises the likelihood with the prediction of the model on a strongly augmented version of $x$. Weak augmentations are also used for the supervised part of the loss. In this context,

$$L(\theta; x, y) = \mathbb{E}_{x_1 \sim weak(x)}[-\log(p_\theta(y|x_1))]$$

and

$$H(\theta; x) = \mathbb{E}_{x_1 \sim weak(x)} \left[ \mathbb{1}[\max_y p_{\hat{\theta}}(y|x_1) > \tau] \mathbb{E}_{x_2 \sim strong(x)}[-\log(p_\theta(\arg\max_y p_{\hat{\theta}}(y|x_1)|x_2))] \right]$$

where $x_1$ is a weak augmentation of $x$ and $x_2$ is a strong augmentation. We tried to debias an implementation of Fixmatch [1] however training was very unstable and led to model that were much worst than the complete case. We believed that this behaviour is because the supervised part of

---

[1]`https://https://github.com/LeeDoYup/FixMatch-pytorch`

the loss does not include strong augmentation. Indeed, our theoretical results encourage to have a strong correlation between $L$ and $H$, therefore including strong augmentations in the supervised term. Moreover, a solid baseline for CIFAR-10 using only labelled data integrated strong augmentations (Cubuk et al., 2020). We modify the implementation, see Code in supplementary materials. Therefore, the supervised loss term can be written as:

$$L(\theta; x, y) = \frac{1}{2} \left( \mathbb{E}_{x_1 \sim weak(x)}[-\log(p_\theta(y|x_1))] + \mathbb{E}_{x_2 \sim strong(x)}[-\log(p_\theta(y|x_2))] \right), \quad (52)$$

where $x_1$ is a weak augmentation of $x$ and $x_2$ is a strong augmentation. This modification encourages us to choose $\lambda = \frac{1}{2}$ as the original Fixmatch implementation used $\lambda = 1$. We also remark that this modification degrades the performance of Fixmatch (less than 2%) reported in the work of Sohn et al. (2020). However, including strong augmentations in the supervised part greatly improves the performance of the Complete Case.

## O  CIFAR and SVHN: Oliver et al. (2018) implementation of consistency-based model.

In this section, we present the results on CIFAR and SVHN by debiasing the implementation of (Oliver et al., 2018) of $\Pi$-Model, Mean-Teacher and VAT [2]. We mimic the experiments of Oliver et al. (2018, figure-4) with the same configuration and the exact same hyperparameters (Oliver et al., 2018, Appendix B and C). We perform an early stopping independently on both cross-entropy and accuracy. As reported below, we reach almost the same results as the biased methods.

### O.1  CIFAR-10

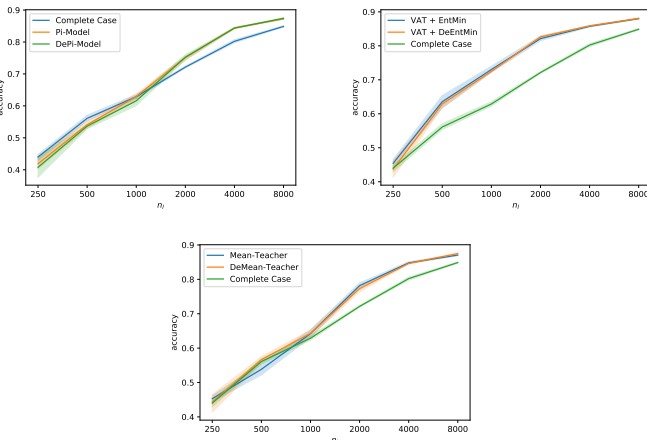

Figure 11: Test accuracy for each SSL approaches on CIFAR-10 with various amounts of labelled data $n_l$.(Left) $\Pi$-model and De$\Pi$-model. (Right) VAT+EntMin and VAT+DeEntMin. (Bottom) Mean-teacher and DeMean-teacher. Shadows represent 95% CI.

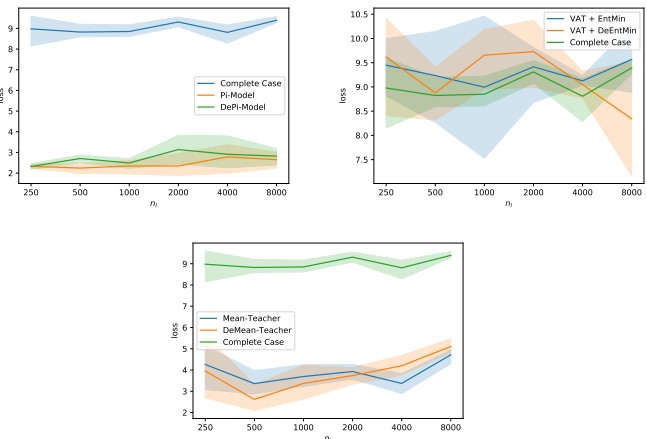

Figure 12: Test cross-entropy for each SSL approaches on CIFAR-10 with various amounts of labelled data $n_l$.(Left) $\Pi$-model and De$\Pi$-model. (Right) VAT+EntMin and VAT+DeEntMin. (Bottom) Mean-teacher and DeMean-teacher. Shadows represent 95% CI.

---

[2] https://github.com/brain-research/realistic-ssl-evaluation

 **O.2   SVHN**

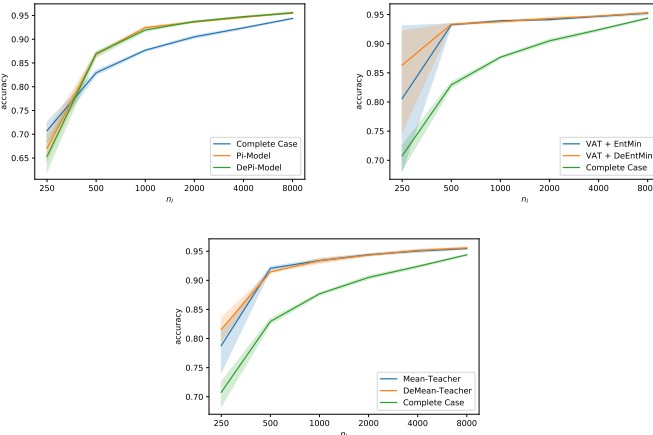

Figure 13: Test accuracy for each SSL approaches on CIFAR-10 with various amounts of labelled data $n_l$.(Left) $\Pi$-model and De$\Pi$-model. (Right) VAT+EntMin and VAT+DeEntMin. (Bottom) Mean-teacher and DeMean-teacher. Shadows represent 95% CI.

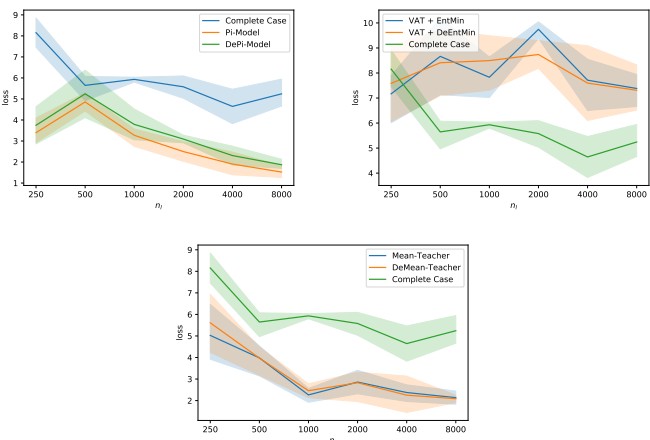

Figure 14: Test cross-entropy for each SSL approaches on CIFAR-10 with various amounts of labelled data $n_l$.(Left) $\Pi$-model and De$\Pi$-model. (Right) VAT+EntMin and VAT+DeEntMin. (Bottom) Mean-teacher and DeMean-teacher. Shadows represent 95% CI.

# P    Tabular benchmarks

In this section, we tested these methods against the benchmarks of Chapelle et al., 2006, Chapter 21 and UCI datasets already used in an SSL context in (Guo et al., 2010). We trained a logistic regression for the case of 100 labelled datapoints and finetune $\lambda$ with a very small validation set, 20 datapoints. We evaluated the performance in accuracy and cross-entropy of PseudoLabel, EntMin, DePseudoLabel and DeEntMin

## P.1    SSL Benchmark

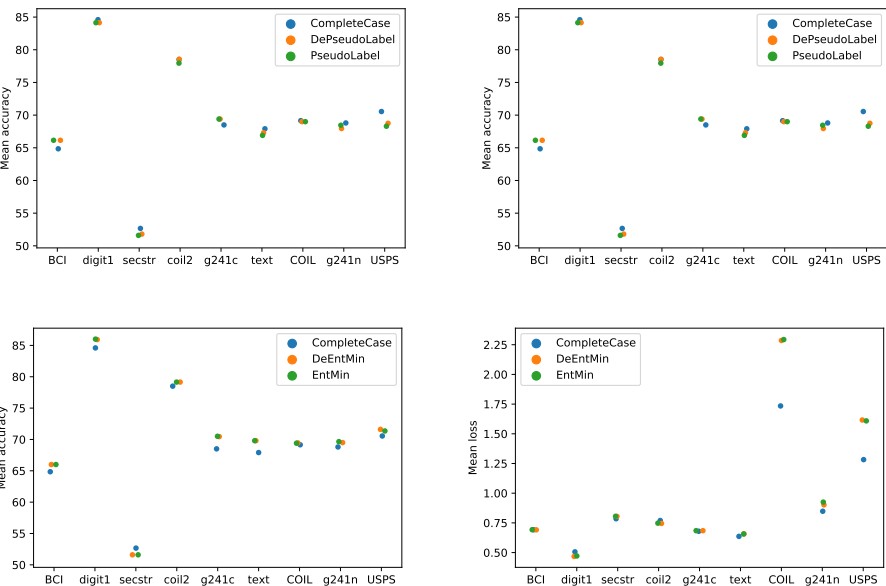

Figure 15: Mean accuracy and cross-entropy for each SSL datasets (Chapelle et al., 2006) on a logistic regression. (Top-Left) PseudoLabel and DePseudoLabel accuracy (Top-Right) PseudoLabel and DePseudoLabel cross-entropy (Bottom-Left) EntMin and DeEntMin accuracy (Bottom-Right) EntMin and DeEntMin cross-entropy.

 **P.2    UCI datasets**

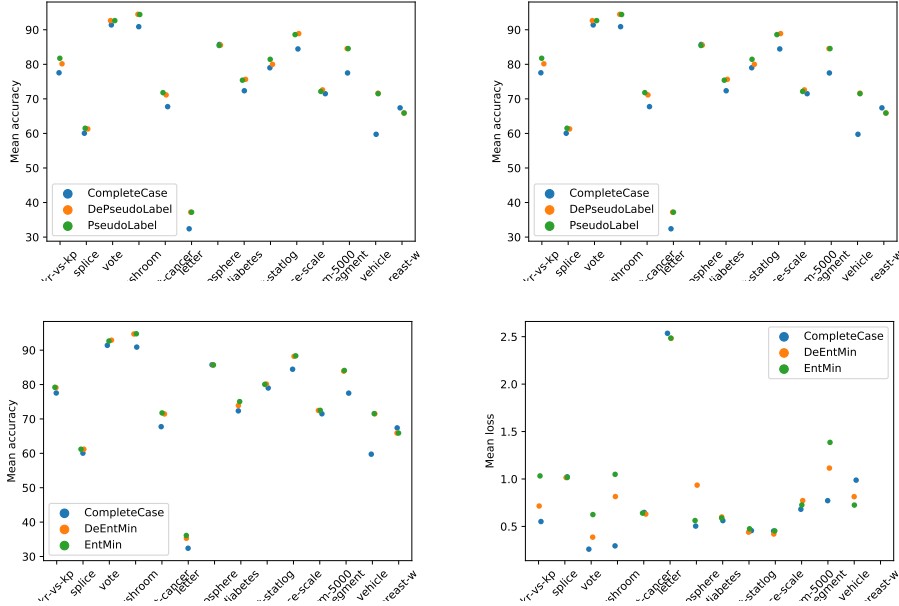

Figure 16: Mean accuracy and cross-entropy for each UCI datasets (Guo et al., 2010) on a logistic regression. (Top-Left) PseudoLabel and DePseudoLabel accuracy (Top-Right) PseudoLabel and DePseudoLabel cross-entropy (Bottom-Left) EntMin and DeEntMin accuracy (Bottom-Right) EntMin and DeEntMin cross-entropy.

# Q    Computation details

## Q.1    Computation resources

Deep Learning experiments of this work required approximately 9,200 hours of GPU computation. In particular, Fixmatch was trained using 4 GPUs. Here are the details:

- MNIST : 300 hours
- medMNIST: 3 hours
- CIFAR-10: 525 hours
- CIFAR-100: 1500 hours
- Fixmatch : 960 hours
- Realistic SSL evaluation on both CIFAR and SVHN: 5880 hours

## Q.2    Computation libraries and tools

- Python (Van Rossum & Drake Jr, 1995)
- PyTorch (Paszke et al., 2019)
- TensorFlow (Abadi et al., 2015)
- Scikit-learn (Pedregosa et al., 2011)
- Seaborn (Waskom et al., 2017)
- Python imaging library (Lundh et al., 2012)
- Numpy (Harris et al., 2020)
- Pandas (McKinney et al., 2010)
- RandAugment (Cubuk et al., 2020)
- Fixmatch-Pytorch [3]
- Realistic-SSL-evaluation (Oliver et al., 2018)

---

[3] `https://https://github.com/LeeDoYup/FixMatch-pytorch`

