# OpenReview forum: "Don’t fear the unlabelled: safe semi-supervised learning via simple debiasing"
_NeurIPS.cc/2022/Conference — NeurIPS 2022 Submitted_

### Official Review · Reviewer_5CzW · 2022-07-09

**Rating:** 6
**Confidence:** 5
**Soundness:** 3 good
**Presentation:** 3 good
**Contribution:** 3 good

**Summary:**

The authors propose a method of removing the bias in semi-supervised learning: debiased SSL (DeSSL). The paper provides generalization error bounds for the proposed methods based on the missing completely at random (MCAR) assumption. The authors also demonstrate the implementation of the debiased component and evaluate debiased versions of different existing SSL methods, such as the Pseudo-label method and Fixmatch, and show that debiasing can compete with classic deep SSL techniques in various settings by providing better calibrated models.

**Questions:**

Since the focus of the proposed method is to reduce bias, is there a way to directly evaluate the bias reduction in addition to the results reported in the experiments?



**Limitations:**

yes

**Strengths And Weaknesses:**

The paper has good clarity on the theory presented and has generally sound approaches and good results. The literature review is comprehensive and leads to the proposed method naturally. Technical results are not particularly complex but flow naturally and results are intuitive.

There are several aspects of the paper that can be strengthened:

The key to the proof of the consistency is the assumption that the true parameter optimizes the proposed "debiased" objective under the true distribution. A natural question is  when such an assumption is viable, and when it is not. For example, would there be situations when the true parameter approximately optimizes the objective without adjusting for bias using the labeled data?

The theory in the paper is built on the assumption of missing completely at random (MCAR), which is often considered in practice a strong condition and sometimes not realistic. Therefore it would be helpful to evaluate how much the results rely on this assumption and provide some intuition on how MCAR is used in the theory.

In problems emphasizing the overall prediction error, it could be argued that the bias-variance trade-off can be more important than the bias itself. This paper partially addresses this issue by finding the optimal $\lambda$ in minimizing the variance of the objective function. The resulting format of the minimal variance is intuitive. An alternative and more direct calculation is to find the $\lambda$ that leads to the smallest asymptotic variance of the $\hat{\theta}$, which should be easy to obtain from the $M$-estimator results.

The experiments section is comprehensive. In evaluate the influence of $\lambda$, it would be helpful to visually mark the estimated optimal $\lambda$ and examine whether it actually achieves better performance.

Minor issues:
Some definitions are missing which may prevent the reader from following the technical results:
eq (1) $n_l$
eq (2) $n_u$
eq (4) several items are not defined

---

> ### Author Response · Authors · 2022-08-02
> **Answer to Reviewer 5CzW**
>
> Many thanks for your comments and assessment of our paper!
>
> >The key to the proof of the consistency is the assumption that the true parameter optimizes the proposed "debiased" objective under the true distribution. A natural question is when such an assumption is viable
>
> The optimal $\theta^*$ is defined as a minimiser of the theoretical risk $\mathcal{R}(\theta)=\mathbb{E}[L(\theta;x,y)]$ and is not directly related to our debiased objective. We never assume the existence of a "true" parameter in the sense of model misspecification. The risk $\mathcal{R}(\theta)$ being intractable, we define an estimator of it that we minimise.
> The key to the consistency Theorem is the unbiasedness of this estimator and then that the optimal parameter is a minimiser of the expectation of our estimator.
>
> > Would there be situations when the true parameter approximately optimizes the objective without adjusting for bias using the labeled data?".
>
> Indeed, $\theta^*$ would minimise the classic SSL estimator if it minimises also the expectation of $H(\theta;x)$. It is likely that this can be the case with sufficiently strong assumption on the data distribution, akin to the cluster assumption.
>
>
> >It would be helpful to evaluate how much the results rely on this assumption and provide some intuition on how MCAR is used in the theory.
>
> MCAR means $r$ is independent of $x$ and $y$. This property is mandatory for all our theoretical results to simplify the expression of expectation and variance. Indeed, we need the assumption to prove that DeSSL is an unbiased estimator of the risk. Also, we remark that without this assumption, the complete case (using only labelled data) is not an unbiased estimator of the risk.
>
>
> > it could be argued that the bias-variance trade-off can be more important than the bias itself.
>
> Classic SSL methods are not really designed to better estimate $\mathcal{R}(\theta)$. On the other hand, DeSSL is based on being a better estimator. Therefore, bias-variance trade-offs are not relevant for non-unbiased SSL as shown in the following equation.
> Indeed, the trade-off is highlighted by the expression of MSE expectation between the quantity to estimate and the estimator as a bias term and variance term. In the case of biased SSL:
> $$
>     \mathbb{E}[(\mathcal{R}(\theta)-\mathcal{R}_{SSL}(\theta))^2]
>      = \lambda^2\mathbb{E}[H(\theta;x)]^2 + \frac{1}{n_l}\mathbb{V}(L(\theta;x,y)) + \lambda^2\mathbb{V}(H(\theta;x))+ \sigma^2
> $$
>
> with $\sigma$ an irreducible error. This means that $\mathcal{R}_{SSL}$ will always be a worse estimator than the complete case. Of course, it does not mean that the SSL methods are worse than the complete case but just that the bias-variance trade-off is not relevant in that context.
>
>
> We thank you for your advice on computing the $\lambda$ that leads to a minimal asymptotic variance! We add a theorem in Appendix I to prove the asymptotic normality $\Sigma_{DeSSL}$ can be optimised with respect to $\lambda$:
>
> $$
>         \lambda_{opt} = (1-\pi)\frac{\mathbf{Tr}(V_{\theta^*}^{-1}K_{\theta^*}V_{\theta^*}^{-1})}{\mathbf{Tr}(V_{\theta^*}^{-1}\mathbb{E}\left[\nabla H(\theta^*;x)\nabla H(\theta^*;x)^T\right]V_{\theta^*}^{-1})},
> $$
>
> where $K_{\theta^*}$ is cross-covariance matrice between  the gradients of $H$ and $L$ and $V_{\theta^*}$ the risk Hessian.
> At this $\lambda$, the asymptotic variance of our estimate is smaller than the complete case.
>
>
> >Is there a way to directly evaluate the bias reduction?
>
> The bias which is equal to $\lambda\mathbb{E}[H(\theta;x)]$ for classic methods, and null for DeSSL. However, we think that looking at standard biased SSL from a Monte Carlo perspective is not very fair to these methods.  Their bias is an increasing function of $\lambda$, so the minimal biased SSL method is then the complete case ($\lambda=0$). Moreover, heavily biased SSL methods can work very well, for instance, Fixmatch. An interesting thought experiment is to take the ideal case where $H(\theta;x) = L(\theta;x,y)$, in this case the expectation of $\mathcal{R}_{SSL}(\theta)$ is proportional but not equal to the true risk. While our method's motivation is linked to Monte Carlo techniques (in particular on the concepts of bias and variance), classic SSL methods do not have these motivations.
>
> >It would be helpful to visually mark the estimated optimal $\lambda$ and examine whether it actually achieves better performance.
>
> As suggested by **Reviewer UPnt**, we estimate $\lambda_{opt}$ on the test set on an already trained model to provide further intuition on the range of $\lambda$ for which we have a variance reduction regime (see Appendix M.2). However, $\lambda_{opt}$ depends of the parameter $\theta$ and then has to be updated during the training at each gradient step. We think that it can be confusing to add $\lambda_{opt}$ on our figures as it is an evolving quantity. However, if you find this information relevant we can add it to our figures during the discussion period.

---

> > ### Comment · Reviewer_5CzW · 2022-08-08
> > **Thank you for your effort**
> >
> > Thanks for replying to my earlier comments and clarifying the questions and issues. I have read carefully your response and think this paper  can be a useful addition to the current studies. I hold my original score of 6 and support this paper to be accepted.

---

> > > ### Author Response · Authors · 2022-08-09
> > > **Follow up to your comment**
> > >
> > > Thank you again for your suggestions and for supporting our paper. The asymptotic normality was a good idea.

---

### Official Review · Reviewer_ZE27 · 2022-07-10

**Rating:** 7
**Confidence:** 4
**Soundness:** 3 good
**Presentation:** 3 good
**Contribution:** 3 good

**Summary:**

This paper proposes to debias the training objective for semi-supervised learning (SSL) methods. The paper suggests a simple modification to the training objective, which de-biases it, by reducing the certainty on labelled datapoints, and considering the whole objective as an optimization of the complete case with a constraint on the two sets of labelled and unlabelled datapoints to satisfy the same property on the SSL loss $H$ on average. Then, the authors proceed to prove statistical learning results on the debiased SSL method, and empirically demonstrate the usefulness of the approach.

**Questions:**

What will happen if we replace the labeled data points used for debiasing the objective with unlabeled datapoints? I assume that the approach would fail, because the two objectives on the unlabelled datapoints will be competing with each other.

Minor:

Line 35: "notion safe semi-supervised learning" <- "notion of safe semi-supervised learning". There are other typos. Please, proofread carefully.

Figure 2: in the caption, it should be "top" and "bottom" instead of "left" and "right". Likewise for Figure 3.

You have missed to put some notations in the equation environment within the appendix, please proofread carefully.


**Limitations:**

I think the paper does not yield any major limitations. However, I don't see a dedicated section about limitations in the main text. I suggest the author spend some time thinking about it.

**Strengths And Weaknesses:**

Strengths:

1) The idea of the paper is clear and sound. The bound on the variance of the DeSSL risk estimate and the generalization bound are useful. They give us generalization insights about SSL, and the benefit from the correlation between $L$ and $H$.

2) The experiments demonstrate that the intuition about the usefulness of debiasing benefits standard SSL benchmarks. I like the toy task example as well.

3) Overall, the paper is clearly written. The proofs seem correct to me, although I have not been able to check every single detail.

Weaknesses:

1) It would be interesting to test your debiasing on a larger task, such as ImageNet. For example, how does debiasing work for PAWS (https://arxiv.org/abs/2104.13963)?

2) I think the paper should discuss the role of the debiasing, and the role of the reducing of the confidence on the labeled datapoints. I think the latter is important for the success of the method, and the theory as well. However, just by looking at the main paper, this doesn't stand out.

For example, we can achieve debiasing by splitting the unlabelled dataset in two parts and using the red term on one part and the blue term on the other (using the color convention from Equation 5 in the main text). Doing, this, we will make the estimate unbiased, but we will not learn good representations (I think), because we have two conflicting objectives on the unlabelled dataset. Having the blue term on the labelled dataset is very important, I think, but it does not stand out in the paper right now. I think the blue term helps with calibration, as the ECE experiments show in the Appendix. Please, discuss this more.

---

> ### Author Response · Authors · 2022-08-02
> **Answer to Reviewer ZE27**
>
>
> Many thanks for your comments and assessment of our paper!
>
> >It would be interesting to test your debiasing on a larger task, such as ImageNet".
>
> Unfortunately, as presented in Appendix O, SSL methods require a lot of computing resources and time to be trained. For instance, Fixmatch's authors said about ImageNet in the issue \#31 of the official GitHub repository "We trained on TPU with 32 cores, which should be roughly equivalent in terms of compute to 32 v100. [...] In our setup it took about 2.5 - 3 days to train for full 3000 epochs.". To add results on a larger task and hard setting, we trained a model on cifar-100 with both Fixmatch and DeFixmatch using only $n_l=400$ labelled datapoints. We show that DeFixmatch outperforms Fixmatch in both accuracy and cross-entropy.
>
>
> >For example, how does debiasing work for PAWS".
> >
> PAWS is somehow close to Fixmatch and UDA in the sense that it aims to minimise a cross-entropy of the prediction between two augmentations of the same datapoint. The difference is that labelled data points are not used to train the model but only to compute a similarity classifier ($\pi_d$ in the paper). Then, their overall objective is not an estimator of a theoretical risk as classic SSL methods can be interpreted. In that sense, we think that applying our DeSSL method to PAWS is not straightforward.
>
> >"Discuss the role of the debiasing and the role of the reducing of the confidence on the labeled datapoints."
>
> Our first motivation for debiasing the objective was to find theoretical results on SSL methods.
> Then, the debiasing term penalises the confidence on the labelled datapoints and then the overfitting on the training dataset. As remarked in paragraph 3.1, between lines 229 and 234, Pereyra et al. [2017] showed that penalising low entropy models acts as a strong regulariser in supervised settings. This comforts the idea of penalising low entropy on the labelled dataset, i.e. debiasing the entropy minimisation with the labelled dataset. Considering pseudo-Label based methods, the objective for the labelled data will be to predict the correct labels with moderate confidence. In this case, the labelled objective is similar to the concept of plausibility inference described by Barndorff-Nielsen [1976].
>
> >What will happen if we replace the labeled data points used for debiasing the objective with unlabeled datapoints?
>
> We can intuitively understand the benefits of debiasing the estimator with labelled data to penalise the confidence of the model on these datapoints. As you rightly suggest, the debiasing can be performed on any subset of the training data. However, in regard to the variance of the estimator, we can prove that debiasing with only the labelled data or the whole dataset are both optimal and equivalent. We add details and the proof in Appendix E.

---

> > ### Comment · Reviewer_ZE27 · 2022-08-04
> > **Thanks for your effort**
> >
> > Thank you for addressing my concerns and suggestions. I particularly like the added discussion about the optimal subset on which to apply debiasing. I think this is an important discussion about making your proposal effective and meaningful. Please, address it in depth in the main text.
> >
> > I appreciate your effort, and I raise my score. I think this paper will be a useful contribution to the community.
> >
> > minor: Please, clean up your math in the appendix. There are some typos.

---

> > > ### Author Response · Authors · 2022-08-05
> > > **Follow up to your comment**
> > >
> > > Thank you again for your suggestion. We think it will add real value to our paper and we will add it to section 3.1 (*Does the DeSSL risk estimator make sense?*) of the camera-ready version of the paper if it is accepted.
> > >
> > > We will proofread the supplementary materials to correct typos and clean up the math.
> > > Thank you for considering raising your score.

---

### Official Review · Reviewer_M2E5 · 2022-07-11

**Rating:** 3
**Confidence:** 4
**Soundness:** 2 fair
**Presentation:** 2 fair
**Contribution:** 2 fair

**Summary:**

This paper focuses on semi-supervised learning and tries to give theoretical guarantee about the safeness of semi-supervised learning methods. Specifically, the authors propose a dibiasing approach, and combine it with the deep SSL methods. Theoretical ananysis about the generalization are discussed.

**Questions:**

1) Why is the proposal guaranteed to be safe? It seems that the results cannot guarantee that the proposal will always be better than the simple supervised learning method.

**Limitations:**

Yes

**Strengths And Weaknesses:**

Strengths:

1) This paper focuses on safe semi-supervised learning problems. This is an important problem for SSL methods, and it is a good attempt to give a theoretical analysis of semi-supervised learning methods.

Weakness

1) The paper's presentation is very poor and very hard to follow. The figures and tables need to be reorganized.

2) The paper gives a new definition of safe SSL: "a SSL algorithm is safe if it has theoretical guarantees that are similar or stronger to the complete case baseline". How to judge whether the theory is stronger?  The "safe" is related to the performance, i.e., the performance is guaranteed not to be reduced compared with the baseline supervised learning method.

3)  The theoretical result does not give a guarantee that the performance will always be better than the supervised learning method. So it does not consist with the author's claim.

---

> ### Author Response · Authors · 2022-08-02
> **Answer to Reviewer M2E5**
>
> Many thanks for your comments and assessment of our paper!
>
> >The paper's presentation is very poor and very hard to follow. The figures and tables need to be reorganized.
>
> Can you detail a bit your concerns regarding the organisation and your understanding of the paper? We would benefit a lot from your comments.
>
>
> >How to judge whether the theory is stronger?
>
> Our results prove that the proposed method provides similar guarantees in terms of calibration and consistency and is stronger in terms of the estimator's variance (here "stronger" means clearly lower variance). Also, the generalisation error bound is similar to the one of the complete case. Following the advice of reviewer 4, we also prove that our method has a better asymptotic variance than the complete case.
>
> >Why is the proposal guaranteed to be safe?
>
> It is unfortunately impossible to ensure performance improvement without strong assumptions about the data distribution. We discuss this point in the first paragraph of part 2.3 in our paper and in Appendix C. For instance, S4VM [Li et al., 2014] ensures better performance using the low-density assumption and considering having access to the true model. On the other hand, Schölkopf et al. [2012] show that depending on the causal relationship between the data and the labels, SSL will always fail.
> Therefore, ensuring stronger or similar theoretical guarantees is an adequate definition of safe for SSL.

---

### Official Review · Reviewer_UPnt · 2022-07-12

**Rating:** 4
**Confidence:** 3
**Soundness:** 3 good
**Presentation:** 2 fair
**Contribution:** 2 fair

**Summary:**

This paper suggests a simple denoising technique for existing semi-supervised learning methods (with surrogate objectives). The proposed objective is shown to have less variance in some cases, and is now consistent when $n->\infty$.

**Questions:**

- The first point in my weaknesses section regarding the theory part.

- If the authors can provide a more complete empirical evaluation of the debiasing term with the most common existing SSL methods (pseudo label, entropy minimization, and consistency-based) to show the benefit of the proposed technique.

**Limitations:**

The authors claim in the checklist that they have discussed the limitations of the work in Section 5, but no limitations are mentioned there.

**Strengths And Weaknesses:**

**Strengths:**
- The debiasing technique is simple and easy to implement, which can be added to existing SSL methods without much effort.
- The authors provide solid theoretical contributions regarding the variance of the proposed objective and the consistency of its minimizer.

**Weaknesses**
- Some missing parts on the theory side. I am more interested in the reduced variance of the proposed objective. The authors showed that at $\lambda_{opt}$, the objective has less variance than the standard complete case. However, this $\lambda_{opt}$ value is often unknown or hard to estimate in practice due to $\rho_{L,H}$, so the theorem is less meaningful. I wonder if we can find out the range of $\lambda$ in which we have reduced/increased variance. For example, if the range for reduced variance is small, the method would be less robust (too sensitive to the choice of $\lambda$), thus becoming less meaningful.

- The experiments section is weak. Why don't the authors verify the proposed debiasing term with all three well-known pseudo label, entropy minimization, and consistency-based methods for all experiments (MNIST, MeMNIST, CIFAR)? I imagine these experiments are not too computationally expensive.

---

> ### Author Response · Authors · 2022-08-02
> **Answer to reviewer UPnt**
>
>
> Many thanks for your comments and assessment of our paper!
>
> >I wonder if we can find out the range of $\lambda$ in which we have reduced/increased variance.
>
> We thank you for this remark that enriches our analysis of the variance reduction theorem. Indeed, the theorem shows two regimes, a reduced variance regime between $0$ and $2\lambda_{opt}$ and an increased variance regime above. To illustrate the range of the reduced variance regime, we estimate $\lambda_{opt}$ on the test set for CIFAR-10 by training a CNN13 using only $4,000$ labelled data on 200 epochs. The value of $\lambda_{opt}$ is 1.67, 31.16, and 0.66 for entropy minimisation, pseudo label and, Fixmatch. Therefore, the reduced variance regime covers the intuitive choices of $\lambda$ in the SSL literature. Unfortunately, computing $\lambda_{opt}$ on the test set is not applicable in practice.
>
> >Why don't the authors verify the proposed debiasing term with all three well-known pseudo-label, entropy minimization, and consistency-based methods for all experiments (MNIST, MeMNIST, CIFAR)?
>
> In our experiment, we remark that entropy minimisation is not efficient in a deep learning setting and often provides no improvement nor degradation for decent $\lambda$. We also want to remind you that SSL experiments are expensive, requiring GPUs and several hours/days for datasets such as CIFAR (see Appendix Q). Therefore, in our experiment, we test the more effective methods for each dataset according to the literature. To add results on a larger task and hard setting, we trained a model on cifar-100 with both Fixmatch and DeFixmatch using only $n_l=400$ labelled datapoints. We show that DeFixmatch outperforms Fixmatch in both accuracy and cross-entropy. We test consistency-based methods such as VAT, $\Pi$-model, and Mean Teacher on CIFAR and SVHN in Appendix O. However if you find interesting any additional experiments, we may do them during the discussion period.

---

### Author Response · Authors · 2022-08-02
**General comments**


We thank the reviewers for the valuable feedback and we appreciate their assessment that ''The paper has good clarity on the theory presented and has generally sound approaches and good results. The literature review is comprehensive and leads to the proposed method naturally'' (**Reviewer 5CzW**) and that ''the paper is clearly written'' (**Reviewer ZE27**). They note that on the more practical side, ''The debiasing technique is simple and easy to implement, which can be added to existing SSL methods without much effort'' (**Reviewer UPnt**) and that ''it is a good attempt to give a theoretical analysis of semi-supervised learning methods'' (**Reviewer M2E5**). Reviewers also noted the ''solid theoretical contributions'' (**Reviewer UPnt**) of our paper and the usefulness of our experiments ''the experiments demonstrate the intuition about the usefulness of debiasing benefits standard SSL benchmarks'' (**Reviewer ZE27**).

Following your remarks, we have made the following modifications to the paper (modifications are in red):
* As asked by **Reviewer 5CzW**, we added a new theorem on the asymptotic normality of $\hat{\theta}_{DeSSL}$ and showed that the asymptotic variance can be optimised with respect to $\lambda$ (see Appendix I).
* **Reviewer ZE27** questioned the choice of debiasing with the labelled dataset. We proved that debiasing with the labelled dataset is optimal compared to using unlabelled data instead. We explain the intuition about penalising the confidence of the model on labelled datapoints (See Appendix F).
* **Reviewer UPnt** and **Reviewer 5CzW** asked for more detail on $\lambda_{opt}$ and the range of variance reduction. We show that the range of variance reduction is $[0,2\lambda_{opt}]$. We compute $\lambda_{opt}$ on the test for CIFAR-10 for Entropy minimisation, Pseudo-Label and Fixmatch and present these values in Appendix M.2.
* **Reviewer UPnt** and **Reviewer ZE27** asked to test the debiasing method on larger tasks. We first remind that SSL are time and power expensive. Indeed, training a model on Cifar-100 with Fixmatch requires 2.7 days on 4GPUs. We added experiments on Fixmatch and DeFixmatch on Cifar-100 in a really hard setting ($n_l=400$, 4 points per class). We remind that we slightly modified the supervised loss of Fixmatch to make it compatible with our setting for Cifar-10 (See Appendix N). While this modified version of Fixmatch worked well for Cifar-10, it did not work properly for Cifar-100. So, we also compared DeFixmatch to the original Fixmatch.  Here are the results. DeFixmatch outperforms Fixmatch in both accuracy and cross-entropy (See Appendix N). Interestingly, our debiasing takes a method that completely fails and turns it into a method close to the state-of-the-art.


|               | Fixmatch | Modified Fixmatch | DeFixmatch |
|:-------------:| -------- |:-----------------:| ----------:|
|   Accuracy    | 41.52    |       1.31        |      43.16 |
| Cross entropy | 6.07     |       6.89        |       5.43 |
* We have made a few small clarifications (see details in the individual responses).

Several reviewers asked us to clarify the limitations of our approach. These were notably mentioned in Section 5 as avenues for future work:
* Going beyond the MCAR assumption which is essential in all our theoretical results.
* We did not succeed to estimate $\lambda$ using the training data according to the formula given in Theorem 3.1, as explained in lines 258 to 260.

We thank you again for the good quality of your comments and remarks and look forward to any additional questions or suggestions.

---

### Meta-Review · Area_Chair_jAoj · 2022-08-27

**Recommendation:** Reject
**Confidence:** Certain

**Metareview:**

This paper provides an interesting generalized perspective on SSL techniques and proposes a debiasing technique that can be viewed as decreasing the variance of the risk estimate. The authors argue that this leads to estimators that are better than the purely supervised estimator under a rather weak assumption called MCAR - which assumes that the probability of a missing label is independent of covariate and label.

Although the exposition and perspective are interesting, this paper is borderline for the following two main reasons:

1. A few reviewers were not convinced of the theoretical result that is supposed to show that unlabeled data strictly helps - indeed, whereas usual variance reduction techniques (e.g for optimization schemes such as SGD etc.) lead to a strict gain in terms of convergence rate, there is no clear asymptotic/high probability statement that indicates statistical gain of the corresponding estimator (which is what we really care about - not the risk estimate). The dependence of lambda_opt on theta (which changes every iteration) does not help in providing such a statement. Since this is the primary contribution of the paper, I would suggest the authors follow through with the analysis to show a gain for the actual estimator compared to the "complete case" (only using supervised data).

On that note, the authors claimed in their rebuttal that they have added an asymptotic variance analysis in Appendix I, which indeed would have made a very valuable point - however, I could only find a copy-pasted version of Theorem 3.1. in Appendix I? Similarly, Appendix F does not seem to include the comparison between debasing using labeled and unlabeled data but instead contains the proof of Theorem 3.2. Perhaps the wrong revision was uploaded, but unfortunately, given the current version, this point is not adequately addressed.

2. If the experimental results were more extensive and conclusive, then the current theorem could have perhaps been alright as a mainly methodological contribution. However, as the authors note, extensive experiments require a lot of compute power - however, given the lack of the ultimate theorem, the methodology becomes the primary contribution and would thus require more experimental evidence as the reviewers asked for.

Addressing one of the above points would push the paper above the acceptance threshold which we hope the authors can pursue in their next submission.

**Award:**

No

---

### Decision · Program_Chairs · 2022-09-14

Reject